# MaRS: A Fast Sampler for Mean Reverting Diffusion Based on ODE and SDE Solvers

**Ao Li** [1,2*]   **Wei Fang** [3,4*]   **Hongbo Zhao** [1,2*]   **Le Lu** [3]   **Ge Yang** [1,2†]   **Minfeng Xu** [3,4†]

[1]School of Artificial Intelligence, University of Chinese Academy of Sciences, Beijing, China
[2]Institute of Automation, Chinese Academy of Sciences (CASIA)
[3]DAMO Academy, Alibaba Group    [4]Hupan Laboratory, Hangzhou, China
{liao2022, zhaohongbo2022, ge.yang}@ia.ac.cn
lucas.fw@alibaba-inc.com  minfengxu@163.com  tiger.lelu@gmail.com

## ABSTRACT

In applications of diffusion models, controllable generation is of practical significance, but is also challenging. Current methods for controllable generation primarily focus on modifying the score function of diffusion models, while Mean Reverting (MR) Diffusion directly modifies the structure of the stochastic differential equation (SDE), making the incorporation of image conditions simpler and more natural. However, current training-free fast samplers are not directly applicable to MR Diffusion. And thus MR Diffusion requires hundreds of NFEs (number of function evaluations) to obtain high-quality samples. In this paper, we propose a new algorithm named MaRS (MR Sampler) to reduce the sampling NFEs of MR Diffusion. We solve the reverse-time SDE and the probability flow ordinary differential equation (PF-ODE) associated with MR Diffusion, and derive semi-analytical solutions. The solutions consist of an analytical function and an integral parameterized by a neural network. Based on this solution, we can generate high-quality samples in fewer steps. Our approach does not require training and supports all mainstream parameterizations, including noise prediction, data prediction and velocity prediction. Extensive experiments demonstrate that MR Sampler maintains high sampling quality with a speedup of 10 to 20 times across ten different image restoration tasks. Our algorithm accelerates the sampling procedure of MR Diffusion, making it more practical in controllable generation. [1]

## 1 INTRODUCTION

Diffusion models have emerged as a powerful class of generative models, demonstrating remarkable capabilities across a variety of applications, including image synthesis (Dhariwal & Nichol, 2021; Ruiz et al., 2023; Rombach et al., 2022) and video generation (Ho et al., 2022a;b). In these applications, controllable generation is very important in practice, but it also poses considerable challenges. Various methods have been proposed to incorporate text or image conditions into the score function of diffusion models(Ho & Salimans, 2022; Ye et al., 2023; Zhang et al., 2023), whereas Mean Reverting (MR) Diffusion offers a new avenue of control in the generation process (Luo et al., 2023b). Previous diffusion models (such as DDPM (Ho et al., 2020)) simulate a diffusion process that gradually transforms data into pure Gaussian noise, followed by learning to reverse this process for sample generation (Song & Ermon, 2020; Song et al., 2021). In contrast, MR Diffusion is designed to produce final states that follow a Gaussian distribution with a non-zero mean, which provides a simple and natural way to introduce image conditions. This characteristic makes MR Diffusion particularly suitable for solving inverse problems and potentially extensible to multi-modal conditions. However, the sampling process of MR Diffusion requires hundreds of iterative steps, which is time-consuming.

---

[*]These authors contributed equally to this work.
[†]Corresponding authors.
[1]Code is available at https://github.com/grrrute/mr-sampler

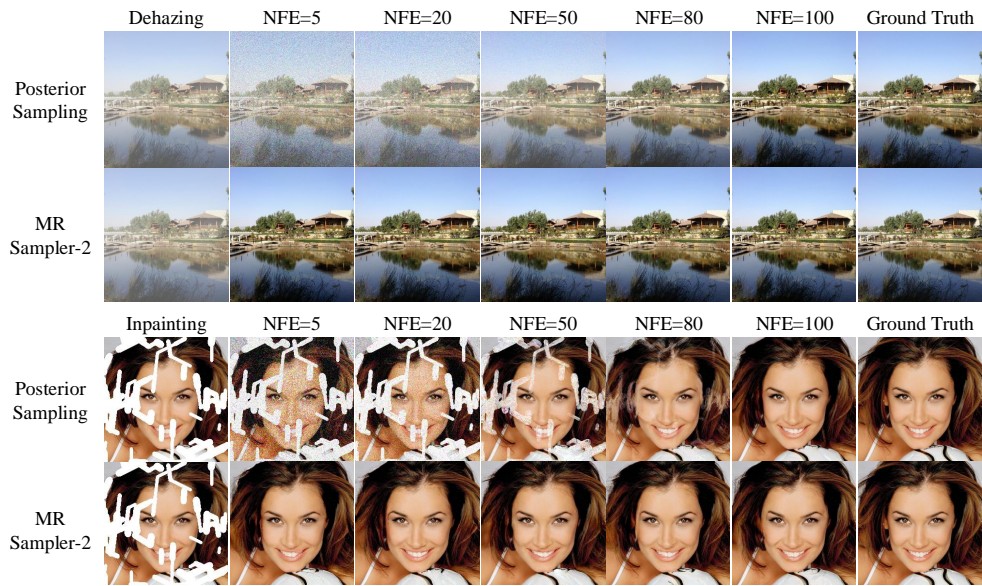

Figure 1: **Qualitative comparisons between MR Sampler and Posterior Sampling.** All images are generated by sampling from a pre-trained MR Diffusion (Luo et al., 2024a) on the RESIDE-6k (Qin et al., 2020b) dataset and the CelebA-HQ (Karras, 2017) dataset.

To improve the sampling efficiency of diffusion models, various acceleration strategies have been proposed, which can be divided into two categories. The first explores methods that establish direct mappings between starting and ending points on the sampling trajectory, enabling acceleration through knowledge distillation (Salimans & Ho, 2022; Song et al., 2023; Liu et al., 2022b). However, such algorithms often come with trade-offs, such as the need for extensive training and limitations in their adaptability across different tasks and datasets. The second category involves the design of fast numerical solvers that increase step sizes while controlling truncation errors, thus allowing for faster convergence to solutions (Lu et al., 2022a; Zhang & Chen, 2022; Song et al., 2020a).

Notably, fast sampling solvers mentioned above are designed for common SDEs such as VPSDE and VESDE (Song et al., 2020b). Due to the difference between these SDEs and MRSDE, existing training-free fast samplers cannot be directly applied to Mean Reverting (MR) Diffusion. In this paper, we propose a novel algorithm named MaRS (MR Sampler) that improves the sampling efficiency of MR Diffusion. Specifically, we solve the reverse-time stochastic differential equation (SDE) and probability flow ordinary differential equation (PF-ODE) (Song et al., 2020b) derived from MRSDE, and obtain a semi-analytical solution, which consists of an analytical function and an integral parameterized by neural networks. We prove that the difference of MRSDE only leads to change in analytical part of solution, which can be calculated precisely. And the integral part can be estimated by discretization methods developed in several previous works (Lu et al., 2022a; Zhang & Chen, 2022; Zhao et al., 2024). We derive sampling formulas for two types of neural network parameterizations: noise prediction (Ho et al., 2020; Song et al., 2020b) and data prediction (Salimans & Ho, 2022). Through theoretical analysis and experimental validation, we demonstrate that data prediction exhibits superior numerical stability compared to noise prediction. Additionally, we propose transformation methods for velocity prediction networks (Salimans & Ho, 2022) so that our algorithm supports all common training objectives. Extensive experiments show that our fast sampler converges in 5 or 10 NFEs with high sampling quality. As illustrated in Figure 1, our algorithm achieves stable performance with speedup factors ranging from 10 to 20.

In summary, our main contributions are as follows:

- We propose *MR Sampler*, a fast sampling algorithm for MR Diffusion, based on solving the PF-ODE and SDE derived from MRSDE. Our algorithm is plug-and-play and can adapt to all common training objectives.

- We demonstrate that posterior sampling (Luo et al., 2024b) for MR Diffusion is equivalent to Euler-Maruyama discretization, whereas MR Sampler computes a semi-analytical solution, thereby eliminating part of approximation errors.
- Through extensive experiments on ten image restoration tasks, we demonstrate that MR Sampler can reduce the required sampling time by a factor of 10 to 20 with comparable sampling quality. Moreover, we reveal that data prediction exhibits superior numerical stability compared to noise prediction.

## 2 BACKGROUND

In this section, we briefly review the basic definitions and characteristics of diffusion probabilistic models and mean-reverting diffusion models.

### 2.1 DIFFUSION PROBABILISTIC MODELS

According to Song et al. (2020b), Diffusion Probabilistic Models (DPMs) can be defined as the solution of the following Itô stochastic differential equation (SDE), which is a stochastic process $\{\boldsymbol{x}_t\}_{t\in[0,T]}$ with $T > 0$, called *forward process*, where $\boldsymbol{x}_t \in \mathbb{R}^D$ is a D-dimensional random variable.

$$\mathrm{d}\boldsymbol{x} = f(\boldsymbol{x}, t)\mathrm{d}t + g(t)\mathrm{d}\boldsymbol{w}. \tag{1}$$

The forward process performs adding noise to the data $\boldsymbol{x}_0$, while there exists a corresponding reverse process that gradually removes the noise and recovers $\boldsymbol{x}_0$. Anderson (1982) shows that the reverse of the forward process is also a solution of an Itô SDE:

$$\mathrm{d}\boldsymbol{x} = [f(\boldsymbol{x}, t) - g(t)^2 \nabla_{\boldsymbol{x}} \log p_t(\boldsymbol{x})]\mathrm{d}t + g(t)\mathrm{d}\bar{\boldsymbol{w}}, \tag{2}$$

where $f$ and $g$ are the drift and diffusion coefficients respectively, $\bar{\boldsymbol{w}}$ is a standard Wiener process running backwards in time, and time $t$ flows from $T$ to $0$, which means $\mathrm{d}t < 0$. The score function $\nabla_{\boldsymbol{x}} \log p_t(\boldsymbol{x})$ is generally intractable and thus a neural network $\boldsymbol{s}_\theta(\boldsymbol{x}, t)$ is used to estimate it by optimizing the following objective (Song et al., 2020b; Hyvärinen & Dayan, 2005):

$$\boldsymbol{\theta}^* = \arg\min_{\boldsymbol{\theta}} \mathbb{E}_t \left\{ \lambda(t) \mathbb{E}_{\boldsymbol{x}_0} \mathbb{E}_{\boldsymbol{x}_t|\boldsymbol{x}_0} \left[ \|\boldsymbol{s}_{\boldsymbol{\theta}}(\boldsymbol{x}_t, t) - \nabla_{\boldsymbol{x}_t} \log p(\boldsymbol{x}_t|\boldsymbol{x}_0)\|_2^2 \right] \right\}. \tag{3}$$

where $\lambda(t) : [0, T] \to \mathbb{R}^+$ is a positive weighting function, $t$ is uniformly sampled over $[0, T]$, $\boldsymbol{x}_0 \sim p_0(\boldsymbol{x})$ and $\boldsymbol{x}_t \sim p(\boldsymbol{x}_t|\boldsymbol{x}_0)$. To facilitate the computation of $p(\boldsymbol{x}_t|\boldsymbol{x}_0)$, the drift coefficient $f(\boldsymbol{x}, t)$ is typically defined as a linear function of $\boldsymbol{x}$, as presented in Eq.(4). Based on the inference by Särkkä & Solin (2019) in Section 5.5, the transition probability $p(\boldsymbol{x}_t|\boldsymbol{x}_0)$ corresponding to Eq.(4) follows Gaussian distribution, as shown in Eq.(5).

$$\mathrm{d}\boldsymbol{x} = f(t)\boldsymbol{x}\mathrm{d}t + g(t)\mathrm{d}\boldsymbol{w}, \tag{4}$$

$$p(\boldsymbol{x}_t|\boldsymbol{x}_0) \sim \mathcal{N}\left( \boldsymbol{x}_t; \boldsymbol{x}_0 e^{\int_0^t f(\tau)\mathrm{d}\tau}, \int_0^t e^{2\int_\tau^t f(\xi)\mathrm{d}\xi} g^2(\tau)\mathrm{d}\tau \cdot \boldsymbol{I} \right). \tag{5}$$

Song et al. (2020b) proved that Denoising Diffusion Probabilistic Models (Ho et al., 2020) and Noise Conditional Score Networks (Song & Ermon, 2019) can be regarded as discretizations of Variance Preserving SDE (VPSDE) and Variance Exploding SDE (VESDE), respectively. As shown in Table 1, the SDEs corresponding to the two most commonly used diffusion models both follow the form of Eq.(4).

Table 1: Two popular SDEs, Variance Preserving SDE (VPSDE) and Variance Exploding SDE (VESDE). $m(t)$ and $v(t)$ refer to mean and variance of the transition probability $p(\boldsymbol{x}_t|\boldsymbol{x}_0)$.

| SDE | $f(t)$ | $g(t)$ | $m(t)$ | $v(t)$ |
|---|---|---|---|---|
| VPSDE(Ho et al., 2020) | $-\frac{1}{2}\beta(t)$ | $\sqrt{\beta(t)}$ | $\boldsymbol{x}_0 e^{-\frac{1}{2}\int_0^t \beta(\tau)\mathrm{d}\tau}$ | $\boldsymbol{I} - \boldsymbol{I} e^{-\int_0^t \beta(\tau)\mathrm{d}\tau}$ |
| VESDE(Song & Ermon, 2019) | $0$ | $\sqrt{\frac{\mathrm{d}[\sigma^2(t)]}{\mathrm{d}t}}$ | $\boldsymbol{x}_0$ | $\left[\sigma^2(t) - \sigma^2(0)\right]\boldsymbol{I}$ |

## 2.2 MEAN REVERTING DIFFUSION MODELS

Luo et al. (2023b) proposed a special case of Itô SDE named Mean Reverting SDE (MRSDE), as follows:

$$\mathrm{d}\boldsymbol{x} = f(t)\left(\boldsymbol{\mu} - \boldsymbol{x}\right)\mathrm{d}t + g(t)\mathrm{d}\boldsymbol{w}, \tag{6}$$

where $\boldsymbol{\mu}$ is a parameter vector that has the same shape of variable $\boldsymbol{x}$, and $f(t), g(t)$ are time-dependent non-negative parameters that control the speed of the mean reversion and stochastic volatility, respectively. To prevent potential confusion, we have substituted the notation used in the original paper (Luo et al., 2023b). For further details, please refer to Appendix B. Under the assumption that $g^2(t)/f(t) = 2\sigma_\infty^2$ for any $t \in [0, T]$ with $T > 0$, Eq.(6) has a closed-form solution, given by

$$\boldsymbol{x}_t = \boldsymbol{x}_0 e^{-\int_0^t f(\tau)\mathrm{d}\tau} + \boldsymbol{\mu}(1 - e^{-\int_0^t f(\tau)\mathrm{d}\tau}) + \sigma_\infty\sqrt{1 - e^{-2\int_0^t f(\tau)\mathrm{d}\tau}}\boldsymbol{z}, \tag{7}$$

where $\sigma_\infty$ is a positive hyper-parameter that determines the standard deviation of $\boldsymbol{x}_t$ when $t \to \infty$ and $\boldsymbol{z} \sim \mathcal{N}(\boldsymbol{0}, \boldsymbol{I})$. Note that $\boldsymbol{x}_t$ starts from $\boldsymbol{x}_0$, and converges to $\boldsymbol{\mu} + \sigma_\infty\boldsymbol{z}$ as $t \to \infty$. According to Anderson (1982)'s result, we can derive the following reverse-time SDE:

$$\mathrm{d}\boldsymbol{x} = \left[ f(t)\left(\boldsymbol{\mu} - \boldsymbol{x}\right) - g^2(t)\nabla_{\boldsymbol{x}} \log p_t(\boldsymbol{x}) \right]\mathrm{d}t + g(t)\mathrm{d}\bar{\boldsymbol{w}}. \tag{8}$$

Similar to DPMs, the score function in Eq.(8) can also be estimated by score matching methods Song & Ermon (2019); Song et al. (2021). Once the score function is known, we can generate $\boldsymbol{x}_0$ from a noisy state $\boldsymbol{x}_T$. In summary, MRSDE illustrates the conversion between two distinct types of data and has demonstrated promising results in image restoration tasks (Luo et al., 2023c).

Various algorithms have been developed to accelerate sampling of VPSDE, including methods like CCDF (Chung et al., 2022), DDIM (Song et al., 2020a), PNDM (Liu et al., 2022a), DPM-Solver (Lu et al., 2022a) and UniPC (Zhao et al., 2024). Additionally, Karras et al. (2022) and Zhou et al. (2024) have introduced techniques for accelerating sampling of VESDE. However, the drift coefficient of VPSDE and VESDE is a linear function of $\boldsymbol{x}$, while the drift coefficient in MRSDE is an affine function w.r.t. $\boldsymbol{x}$, adding an intercept $\boldsymbol{\mu}$ (see Eq.(4) and Eq.(6)). Therefore, current sampling acceleration algorithms cannot be applied to MR Diffusion. To the best of our knowledge, MR Sampler has been the first sampling acceleration algorithm for MR Diffusion so far.

## 3 FAST SAMPLERS FOR MEAN REVERTING DIFFUSION WITH NOISE PREDICTION

According to Song et al. (2020b), the states $\boldsymbol{x}_t$ in the sampling procedure of diffusion models correspond to solutions of reverse-time SDE and PF-ODE. Therefore, we look for ways to accelerate sampling by studying these solutions. In this section, we solve the noise-prediction-based reverse-time SDE and PF-ODE, and we numerically estimate the non-closed-form component of the solution, which serves to accelerate the sampling process of MR diffusion models. Next, we analyze the sampling method currently used by MR Diffusion and demonstrate that this method corresponds to a variant of discretization for the reverse-time MRSDE.

### 3.1 SOLUTIONS TO MEAN REVERTING SDES WITH NOISE PREDICTION

Ho et al. (2020) reported that score matching can be simplified to predicting noise, and Song et al. (2020b) revealed the connection between score function and noise prediction models, which is

$$\nabla_{\boldsymbol{x}_t} \log p(\boldsymbol{x}_t|\boldsymbol{x}_0) = -\frac{\boldsymbol{\epsilon}_\theta(\boldsymbol{x}_t, \boldsymbol{\mu}, t)}{\sigma_t}, \tag{9}$$

where $\sigma_t = \sigma_\infty\sqrt{1 - e^{-2\int_0^t f(\tau)\mathrm{d}\tau}}$ is the standard deviation of the transition distribution $p(\boldsymbol{x}_t|\boldsymbol{x}_0)$. Because $\boldsymbol{\mu}$ is independent of $t$ and $\boldsymbol{x}$, we substitute $\boldsymbol{\epsilon}_\theta(\boldsymbol{x}_t, \boldsymbol{\mu}, t)$ with $\boldsymbol{\epsilon}_\theta(\boldsymbol{x}_t, t)$ for notation simplicity. According to Eq.(9), we can rewrite Eq.(8) as

$$\mathrm{d}\boldsymbol{x} = \left[ f(t)\left(\boldsymbol{\mu} - \boldsymbol{x}\right) + \frac{g^2(t)}{\sigma_t}\boldsymbol{\epsilon}_\theta(\boldsymbol{x}_t, t) \right]\mathrm{d}t + g(t)\mathrm{d}\bar{\boldsymbol{w}}. \tag{10}$$

Using Itô's formula (in the differential form), we can obtain the following semi-analytical solution:

**Proposition 1.** Given an initial value $\boldsymbol{x}_s$ at time $s \in [0, T]$, the solution $\boldsymbol{x}_t$ at time $t \in [0, s]$ of Eq.(10) is

$$\boldsymbol{x}_t = \frac{\alpha_t}{\alpha_s}\boldsymbol{x}_s + \left(1 - \frac{\alpha_t}{\alpha_s}\right)\boldsymbol{\mu} + \alpha_t \int_s^t g^2(\tau)\frac{\boldsymbol{\epsilon}_\theta(\boldsymbol{x}_\tau, \tau)}{\alpha_\tau \sigma_\tau}\mathrm{d}\tau + \sqrt{-\int_s^t \frac{\alpha_t^2}{\alpha_\tau^2}g^2(\tau)\mathrm{d}\tau}\,\boldsymbol{z}, \quad (11)$$

where we denote $\alpha_t := e^{-\int_0^t f(\tau)\mathrm{d}\tau}$ and $\boldsymbol{z} \sim \mathcal{N}(\boldsymbol{0}, \boldsymbol{I})$. The proof is in Appendix A.1.

However, the integral with respect to neural network output is still complicated. There have been several methods (Lu et al., 2022a; Zhang & Chen, 2022; Zhao et al., 2024) to estimate the integral numerically. We follow Lu et al. (2022b)'s method and introduce the half log-SNR $\lambda_t := \log(\alpha_t/\sigma_t)$. Since both $f(t)$ and $g(t)$ are deliberately designed to ensure that $\alpha_t$ is monotonically decreasing over $t$ and $\sigma_t$ is monotonically increasing over $t$. Thus, $\lambda_t$ is a strictly decreasing function of $t$ and there exists an inverse function $t(\lambda)$. Then we can rewrite $g(\tau)$ in Eq.(11) as

$$g^2(\tau) = 2\sigma_\infty^2 f(\tau) = 2f(\tau)(\sigma_\tau^2 + \sigma_\infty^2\alpha_\tau^2) = 2\sigma_\tau^2(f(\tau) + \frac{f(\tau)\sigma_\infty^2\alpha_\tau^2}{\sigma_\tau^2})$$

$$= 2\sigma_\tau^2(f(\tau) + \frac{1}{2\sigma_\tau^2}\frac{\mathrm{d}\sigma_\tau^2}{\mathrm{d}\tau}) = -2\sigma_\tau^2\frac{\mathrm{d}\lambda_\tau}{\mathrm{d}\tau}. \quad (12)$$

By substituting Eq.(12) into Eq.(11), we obtain

$$\boldsymbol{x}_t = \frac{\alpha_t}{\alpha_s}\boldsymbol{x}_s + \left(1 - \frac{\alpha_t}{\alpha_s}\right)\boldsymbol{\mu} - 2\alpha_t \int_{\lambda_s}^{\lambda_t} e^{-\lambda}\boldsymbol{\epsilon}_\theta(\boldsymbol{x}_\lambda, \lambda)\mathrm{d}\lambda + \sigma_t\sqrt{(e^{2(\lambda_t - \lambda_s)} - 1)}\,\boldsymbol{z}, \quad (13)$$

where $\boldsymbol{x}_\lambda := \boldsymbol{x}_{t(\lambda_\tau)}$, $\boldsymbol{\epsilon}_\theta(\boldsymbol{x}_\lambda, \lambda) := \boldsymbol{\epsilon}_\theta(\boldsymbol{x}_{t(\lambda_\tau)}, t(\lambda_\tau))$. According to the methods of exponential integrators (Hochbruck & Ostermann, 2010; 2005), the $(k-1)$-th order Taylor expansion of $\boldsymbol{\epsilon}_\theta(x_\lambda, \lambda)$ and integration-by-parts of the integral part in Eq.(13) yields

$$-2\alpha_t \int_{\lambda_s}^{\lambda_t} e^{-\lambda}\boldsymbol{\epsilon}_\theta(x_\lambda, \lambda)\mathrm{d}\lambda = -2\sigma_t \sum_{n=0}^{k-1}\left[\boldsymbol{\epsilon}_\theta^{(n)}(\boldsymbol{x}_{\lambda_s}, \lambda_s)\left(e^h - \sum_{m=0}^n \frac{(h)^m}{m!}\right)\right] + \mathcal{O}(h^{k+1}), \quad (14)$$

where $h := \lambda_t - \lambda_s$. We drop the discretization error term $\mathcal{O}(h^{k+1})$ and estimate the derivatives with *backward difference method*. We name this algorithm as *MR Sampler-SDE-n-k*, where *n* means noise prediction and *k* is the order. We present details in Algorithm 1 and 2.

## 3.2 SOLUTIONS TO MEAN REVERTING ODEs WITH NOISE PREDICTION

Song et al. (2020b) have illustrated that for any Itô SDE, there exists a *probability flow* ODE, sharing the same marginal distribution $p_t(\boldsymbol{x})$ as a reverse-time SDE. Therefore, the solutions of PF-ODEs are also helpful in acceleration of sampling. Specifically, the PF-ODE corresponding to Eq.(10) is

$$\frac{\mathrm{d}\boldsymbol{x}}{\mathrm{d}t} = f(t)\left(\boldsymbol{\mu} - \boldsymbol{x}\right) + \frac{g^2(t)}{2\sigma_t}\boldsymbol{\epsilon}_\theta(\boldsymbol{x}_t, t). \quad (15)$$

The aforementioned equation exhibits a semi-linear structure with respect to $\boldsymbol{x}$, thus permitting resolution through the method of "variation of constants". We can draw the following conclusions:

**Proposition 2.** Given an initial value $\boldsymbol{x}_s$ at time $s \in [0, T]$, the solution $\boldsymbol{x}_t$ at time $t \in [0, s]$ of Eq.(15) is

$$\boldsymbol{x}_t = \frac{\alpha_t}{\alpha_s}\boldsymbol{x}_s + \left(1 - \frac{\alpha_t}{\alpha_s}\right)\boldsymbol{\mu} + \alpha_t \int_s^t \frac{g^2(\tau)}{2\alpha_\tau \sigma_\tau}\boldsymbol{\epsilon}_\theta(\boldsymbol{x}_\tau, \tau)\mathrm{d}\tau, \quad (16)$$

where $\alpha_t := e^{-\int_0^t f(\tau)\mathrm{d}\tau}$. The proof is in Appendix A.1.

Then we follow the variable substitution and Eq.(12-14) in Section 3.1, and we obtain

$$\boldsymbol{x}_t = \frac{\alpha_t}{\alpha_s}\boldsymbol{x}_s + \left(1 - \frac{\alpha_t}{\alpha_s}\right)\boldsymbol{\mu} - \sigma_t \sum_{n=0}^{k-1}\left[\boldsymbol{\epsilon}_\theta^{(n)}(\boldsymbol{x}_{\lambda_s}, \lambda_s)\left(e^h - \sum_{m=0}^n \frac{(h)^m}{m!}\right)\right] + \mathcal{O}(h^{k+1}), \quad (17)$$

where $\boldsymbol{\epsilon}_\theta^{(n)}(\boldsymbol{x}_\lambda, \lambda) := \frac{\mathrm{d}^n\boldsymbol{\epsilon}_\theta(\boldsymbol{x}_\lambda, \lambda)}{\mathrm{d}\lambda^n}$ is the $n$-th order total derivatives of $\boldsymbol{\epsilon}_\theta$ with respect to $\lambda$. By dropping the discretization error term $\mathcal{O}(h^{k+1})$ and estimating the derivatives of $\boldsymbol{\epsilon}_\theta(\boldsymbol{x}_{\lambda_s}, \lambda_s)$ with *backward difference method*, we design the sampling algorithm from the perspective of ODE (see Algorithm 3 and 4).

### 3.3 POSTERIOR SAMPLING FOR MEAN REVERTING DIFFUSION MODELS

In order to improve the sampling process of Mean Reverting Diffusion, Luo et al. (2024b) proposed the *posterior sampling* algorithm. They define a monotonically increasing time series $\{t_i\}_{i=0}^T$ and the reverse process as a Markov chain:

$$p(\boldsymbol{x}_{1:T} \mid \boldsymbol{x}_0) = p(\boldsymbol{x}_T \mid \boldsymbol{x}_0) \prod_{i=2}^{T} p(\boldsymbol{x}_{i-1} \mid \boldsymbol{x}_i, \boldsymbol{x}_0) \text{ and } \boldsymbol{x}_T \sim \mathcal{N}(\boldsymbol{0}, \boldsymbol{I}), \tag{18}$$

where we denote $\boldsymbol{x}_i := \boldsymbol{x}_{t_i}$ for simplicity. They obtain an optimal posterior distribution by minimizing the negative log-likelihood, which is a Gaussian distribution given by

$$p(\boldsymbol{x}_{i-1} \mid \boldsymbol{x}_i, \boldsymbol{x}_0) = \mathcal{N}(\boldsymbol{x}_{i-1} \mid \tilde{\boldsymbol{\mu}}_i(\boldsymbol{x}_i, \boldsymbol{x}_0), \tilde{\beta}_i \boldsymbol{I}),$$

$$\tilde{\boldsymbol{\mu}}_i(\boldsymbol{x}_i, \boldsymbol{x}_0) = \frac{(1 - \alpha_{i-1}^2)\alpha_i}{(1 - \alpha_i^2)\alpha_{i-1}}(\boldsymbol{x}_i - \boldsymbol{\mu}) + \frac{1 - \frac{\alpha_i^2}{\alpha_{i-1}^2}}{1 - \alpha_i^2}\alpha_{i-1}(\boldsymbol{x}_0 - \boldsymbol{\mu}) + \boldsymbol{\mu}, \tag{19}$$

$$\tilde{\beta}_i = \frac{(1 - \alpha_{i-1}^2)(1 - \frac{\alpha_i^2}{\alpha_{i-1}^2})}{1 - \alpha_i^2},$$

where $\alpha_i = e^{-\int_0^i f(\tau)\mathrm{d}\tau}$ and $\boldsymbol{x}_0 = (\boldsymbol{x}_i - \boldsymbol{\mu} - \sigma_i \boldsymbol{\epsilon}_\theta(\boldsymbol{x}_i, \boldsymbol{\mu}, t_i))/\alpha_i + \boldsymbol{\mu}$. Actually, the reparameterization of posterior distribution in Eq.(19) is equivalent to a variant of the Euler-Maruyama discretization of the reverse-time SDE (see details in Appendix A.2). Specifically, the Euler-Maruyama method computes the solution in the following form:

$$\boldsymbol{x}_t = \boldsymbol{x}_s + \int_s^t \left[ f(\tau)(\boldsymbol{\mu} - \boldsymbol{x}_\tau) + \frac{g^2(\tau)}{\sigma_\tau}\boldsymbol{\epsilon}_\theta(\boldsymbol{x}_\tau, \tau) \right] \mathrm{d}\tau + \int_s^t g(\tau)\mathrm{d}\bar{\boldsymbol{w}}_\tau, \tag{20}$$

which introduces approximation errors from both the analytical term and the non-linear component associated with neural network predictions. In contrast, our approach delivers an exact solution for the analytical part, leading to reduced approximation errors and a higher order of convergence.

## 4 FAST SAMPLERS FOR MEAN REVERTING DIFFUSION WITH DATA PREDICTION

Unfortunately, the sampler based on noise prediction can exhibit substantial instability, particularly with small NFEs, and may perform even worse than *posterior sampling*. It is well recognized that the Taylor expansion has a limited convergence domain, primarily influenced by the derivatives of the neural networks. In fact, higher-order derivatives often result in smaller convergence radii. During the training phase, the noise prediction neural network is designed to fit normally distributed Gaussian noise. When the standard deviation of this Gaussian noise is set to 1, the values of samples can fall outside the range of $[-1, 1]$ with a probability of 34.74%. This discrepancy results in numerical instability in the output of the neural network, causing its derivatives to exhibit more pronounced fluctuations (refer to the experimental results in Section 5 for further details). Consequently, the numerical instability leads to very narrow convergence domains, or in extreme cases, no convergence at all, which ultimately yields awful sampling results.

Lu et al. (2022b) have identified that the choice of parameterization for either ODEs or SDEs is critical for the boundedness of the convergent solution. In contrast to noise prediction, the data prediction model (Salimans & Ho, 2022) focuses on fitting $\boldsymbol{x}_0$, ensuring that its output remains strictly confined within the bounds of $[-1, 1]$, thereby achieving high numerical stability.

### 4.1 SOLUTIONS TO MEAN REVERTING SDES WITH DATA PREDICTION

According to Eq.(7), we can parameterize $\boldsymbol{x}_0$ as follows:

$$\boldsymbol{\epsilon}_\theta(\boldsymbol{x}_t, t) = \frac{\boldsymbol{x}_t - \alpha_t \boldsymbol{x}_\theta(\boldsymbol{x}_t, t) - (1 - \alpha_t)\boldsymbol{\mu}}{\sigma_t}. \tag{21}$$

By substituting Eq.(21) into Eq.(10), we derive the following SDE that incorporates data prediction:

$$d\boldsymbol{x} = \left[ \left( \frac{g^2(t)}{\sigma_t^2} - f(t) \right) \boldsymbol{x} + \left( f(t) - \frac{g^2(t)}{\sigma_t^2}(1 - \alpha_t) \right) \boldsymbol{\mu} - \frac{g^2(t)}{\sigma_t^2} \alpha_t \boldsymbol{x}_\theta(\boldsymbol{x}_t, t) \right] dt + g(t) d\bar{\boldsymbol{w}}. \quad (22)$$

This equation remains semi-linear with respect to $\boldsymbol{x}$ and thus we can employ Itô's formula (in the differential form) to obtain the solution to Eq.(22).

**Proposition 3.** Given an initial value $\boldsymbol{x}_s$ at time $s \in [0, T]$, the solution $\boldsymbol{x}_t$ at time $t \in [0, s]$ of Eq.(22) is

$$\boldsymbol{x}_t = \frac{\sigma_t}{\sigma_s} e^{-(\lambda_t - \lambda_s)} \boldsymbol{x}_s + \boldsymbol{\mu} \left( 1 - \frac{\alpha_t}{\alpha_s} e^{-2(\lambda_t - \lambda_s)} - \alpha_t + \alpha_t e^{-2(\lambda_t - \lambda_s)} \right)$$
$$+ 2\alpha_t \int_{\lambda_s}^{\lambda_t} e^{-2(\lambda_t - \lambda)} \boldsymbol{x}_\theta(\boldsymbol{x}_\lambda, \lambda) d\lambda + \sigma_t \sqrt{1 - e^{-2(\lambda_t - \lambda_s)}} \boldsymbol{z}, \quad (23)$$

where $\boldsymbol{z} \sim \mathcal{N}(\boldsymbol{0}, \boldsymbol{I})$. The proof is in Appendix A.1.

Then we apply Taylor expansion and integration-by-parts to estimate the integral part in Eq.(23) and obtain the stochastic sampling algorithm for data prediction (see details in Algorithm 5 and 6).

## 4.2 SOLUTIONS TO MEAN REVERTING ODEs WITH DATA PREDICTION

By substituting Eq.(21) into Eq.(15), we can obtain the following ODE parameterized by data prediction.

$$\frac{d\boldsymbol{x}}{dt} = \left[ \frac{g^2(t)}{2\sigma_t^2} - f(t) \right] \boldsymbol{x} + \left[ f(t) - \frac{g^2(t)}{2\sigma_t^2}(1 - \alpha_t) \right] \boldsymbol{\mu} - \frac{g^2(t)}{2\sigma_t^2} \alpha_t \boldsymbol{x}_\theta(\boldsymbol{x}_t, t). \quad (24)$$

The incorporation of the parameter $\boldsymbol{\mu}$ does not disrupt the semi-linear structure of the equation with respect to $\boldsymbol{x}$, and $\boldsymbol{\mu}$ is not coupled to the neural network. This implies that analytical part of solutions can still be derived concerning both $\boldsymbol{x}$ and $\boldsymbol{\mu}$. We present the solution below (see Appendix A.1 for a detailed derivation).

**Proposition 4.** Given an initial value $\boldsymbol{x}_s$ at time $s \in [0, T]$, the solution $\boldsymbol{x}_t$ at time $t \in [0, s]$ of Eq.(24) is

$$\boldsymbol{x}_t = \frac{\sigma_t}{\sigma_s} \boldsymbol{x}_s + \boldsymbol{\mu} \left( 1 - \frac{\sigma_t}{\sigma_s} + \frac{\sigma_t}{\sigma_s} \alpha_s - \alpha_t \right) + \sigma_t \int_{\lambda_s}^{\lambda_t} e^\lambda \boldsymbol{x}_\theta(\boldsymbol{x}_\lambda, \lambda) d\lambda. \quad (25)$$

Similarly, only the neural network component requires approximation through the exponential integrator method (Hochbruck & Ostermann, 2005; 2010). And we can obtain the deterministic sampling algorithm for data prediction (see Algorithm 7 and 8 for details).

## 4.3 TRANSFORMATION BETWEEN THREE KINDS OF PARAMETERIZATIONS

There are three mainstream parameterization methods. Ho et al. (2020) introduced a training objective based on noise prediction, while Salimans & Ho (2022) proposed parameterization strategies for data and velocity prediction to keep network outputs stable under the variation of time or log-SNR. All three methods can be regarded as score matching approaches (Song et al., 2020b; Hyvärinen & Dayan, 2005) with weighted coefficients. To ensure our proposed algorithm is compatible with these parameterization strategies, it is necessary to provide transformation formulas for each pairs among the three strategies.

The transformation formula between noise prediction and data prediction can be easily derived from Eq.(7):

$$\begin{cases} \boldsymbol{x}_\theta(t) = \frac{\boldsymbol{x}_t - (1 - \alpha_t)\boldsymbol{\mu} - \sigma_t \boldsymbol{\epsilon}_\theta(t)}{\alpha_t}, \\ \boldsymbol{\epsilon}_\theta(t) = \frac{\boldsymbol{x}_t - \alpha_t \boldsymbol{x}_\theta(t) - (1 - \alpha_t)\boldsymbol{\mu}}{\sigma_t}. \end{cases} \quad (26)$$

For velocity prediction, we define $\phi_t := \arctan(\frac{\sigma_t}{\sigma_\infty \alpha_t})$, which is slightly different from the definition of Salimans & Ho (2022). Then we have $\alpha_t = \cos\phi_t$, $\sigma_t = \sigma_\infty \sin\phi_t$ and hence $\boldsymbol{x}_t = \boldsymbol{x}_0 \cos\phi_t + \boldsymbol{\mu}(1 - \cos\phi_t) + \sigma_\infty \sin(\phi_t)\boldsymbol{\epsilon}$. And the definition of $\boldsymbol{v}(t)$ is

$$\boldsymbol{v}_t = \frac{d\boldsymbol{x}_t}{d\phi_t} = \boldsymbol{\mu}\sin\phi_t - \boldsymbol{x}_0 \sin\phi_t + \sigma_\infty \cos(\phi_t)\boldsymbol{\epsilon}. \quad (27)$$

If we have a score function model $\boldsymbol{v}_\theta(t)$ trained with velocity prediction, we can obtain $\boldsymbol{x}_\theta(t)$ and $\boldsymbol{\epsilon}_\theta(t)$ by (see Appendix A.3 for detailed derivations)

$$\boldsymbol{x}_\theta(t) = \boldsymbol{x}_t \cos\phi_t + \boldsymbol{\mu}(1 - \cos\phi_t) - \boldsymbol{v}_\theta(t)\sin\phi_t, \tag{28}$$

$$\boldsymbol{\epsilon}_\theta(t) = (\boldsymbol{v}_\theta(t)\cos\phi_t + \boldsymbol{x}_t \sin\phi_t - \boldsymbol{\mu}\sin\phi_t)/\sigma_\infty. \tag{29}$$

## 5 EXPERIMENTS

In this section, we conduct extensive experiments to show that MR Sampler can significantly speed up the sampling of existing MR Diffusion. To rigorously validate the effectiveness of our method, we follow the settings and checkpoints from Luo et al. (2024a) and only modify the sampling part. Our experiment is divided into three parts. Section 5.1 compares the sampling results for different NFE cases. Section 5.2 studies the effects of different parameter settings on our algorithm, including network parameterizations and solver types. In Section 5.3, we visualize the sampling trajectories to show the speedup achieved by MR Sampler and analyze why noise prediction gets obviously worse when NFE is less than 20.

### 5.1 MAIN RESULTS

Following Luo et al. (2024a), we conduct experiments with ten different types of image degradation: blurry, hazy, JPEG-compression, low-light, noisy, raindrop, rainy, shadowed, snowy, and inpainting (see Appendix D.1 for details). We adopt LPIPS (Zhang et al., 2018) and FID (Heusel et al., 2017) as main metrics for perceptual evaluation, and also report PSNR and SSIM (Wang et al., 2004) for reference. We compare MR Sampler with other sampling methods, including posterior sampling (Luo et al., 2024b) and Euler-Maruyama discretization (Kloeden et al., 1992). We take two tasks as examples and the metrics are shown in Figure 2. Unless explicitly mentioned, we always use MR Sampler based on SDE solver, with data prediction and uniform $\lambda$. The complete experimental results can be found in Appendix D.3. The results demonstrate that MR Sampler converges in a few (5 or 10) steps and produces samples with stable quality. Our algorithm significantly reduces the time cost without compromising sampling performance, which is of great practical value for MR Diffusion.

### 5.2 EFFECTS OF PARAMETER CHOICE

In Table 2, we compare the results of two network parameterizations. The data prediction shows stable performance across different NFEs. The noise prediction performs similarly to data prediction with large NFEs, but its performance deteriorates significantly with smaller NFEs. The detailed analysis can be found in Section 5.3. In Table 3, we compare MR Sampler-ODE-d-2 and MR Sampler-SDE-d-2 on the *inpainting* task, which are derived from PF-ODE and reverse-time SDE respectively. SDE-based solver works better with a large NFE, whereas ODE-based solver is more effective with a small NFE. In general, neither solver type is inherently better.

Table 2: Ablation study of network parameterizations on the Rain100H dataset.

| NFE | Parameterization | LPIPS↓ | FID↓ | PSNR↑ | SSIM↑ |
|---|---|---|---|---|---|
| 50 | Noise Prediction | **0.0606** | **27.28** | **28.89** | **0.8615** |
| | Data Prediction | 0.0620 | 27.65 | 28.85 | 0.8602 |
| 20 | Noise Prediction | 0.1429 | 47.31 | 27.68 | 0.7954 |
| | Data Prediction | **0.0635** | **27.79** | **28.60** | **0.8559** |
| 10 | Noise Prediction | 1.376 | 402.3 | 6.623 | 0.0114 |
| | Data Prediction | **0.0678** | **29.54** | **28.09** | **0.8483** |
| 5 | Noise Prediction | 1.416 | 447.0 | 5.755 | 0.0051 |
| | Data Prediction | **0.0637** | **26.92** | **28.82** | **0.8685** |

Table 3: Ablation study of solver types on the CelebA-HQ dataset.

| NFE | Solver Type | LPIPS↓ | FID↓ | PSNR↑ | SSIM↑ |
|---|---|---|---|---|---|
| 50 | ODE | 0.0499 | 22.91 | 28.49 | 0.8921 |
| | SDE | **0.0402** | **19.09** | **29.15** | **0.9046** |
| 20 | ODE | 0.0475 | 21.35 | 28.51 | 0.8940 |
| | SDE | **0.0408** | **19.13** | **28.98** | **0.9032** |
| 10 | ODE | **0.0417** | 19.44 | **28.94** | **0.9048** |
| | SDE | 0.0437 | **19.29** | 28.48 | 0.8996 |
| 5 | ODE | **0.0526** | 27.44 | **31.02** | **0.9335** |
| | SDE | 0.0529 | **24.02** | 28.35 | 0.8930 |

### 5.3 ANALYSIS

**Sampling trajectory.** Inspired by the design idea of NCSN (Song & Ermon, 2019), we provide a new perspective of diffusion sampling process. Song & Ermon (2019) consider each data point

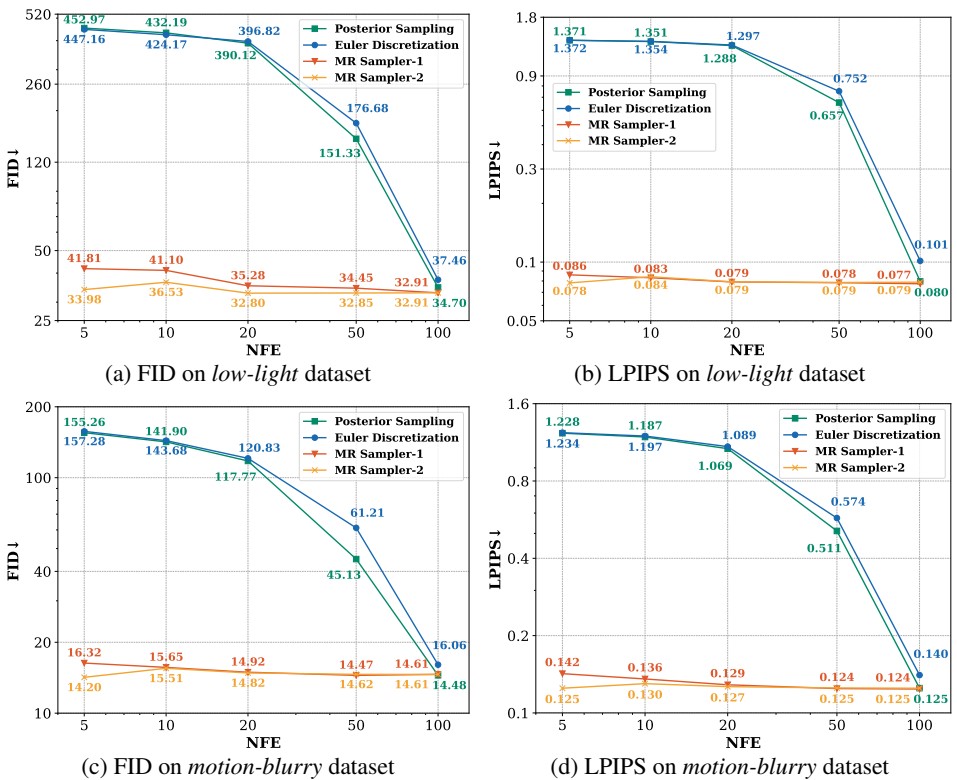

Figure 2: **Perceptual evaluations on *low-light* and *motion-blurry* datasets.**

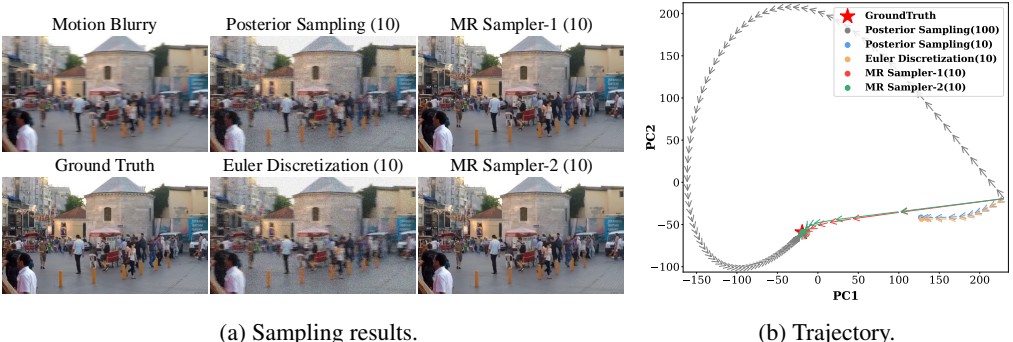

(a) Sampling results.

(b) Trajectory.

Figure 3: **Sampling trajectories.** In (a), we compare our method (with order 1 and order 2) and previous sampling methods (i.e., posterior sampling and Euler discretization) on a motion blurry image. The numbers in parentheses indicate the NFE. In (b), we illustrate trajectories of each sampling method. Previous methods need to take many unnecessary paths to converge. With few NFEs, they fail to reach the ground truth (i.e., the location of $x_0$). Our methods follow a more direct trajectory.

(e.g., an image) as a point in high-dimensional space. During the diffusion process, noise is added to each point $x_0$, causing it to spread throughout the space, while the score function (a neural network) *remembers* the direction towards $x_0$. In the sampling process, we start from a random point by sampling a Gaussian distribution and follow the guidance of the reverse-time SDE (or PF-ODE) and the score function to locate $x_0$. By connecting each intermediate state $x_t$, we obtain a sampling trajectory. However, this trajectory exists in a high-dimensional space, making it difficult to visualize. Therefore, we use Principal Component Analysis (PCA) to reduce $x_t$ to two dimensions, obtaining the projection of the sampling trajectory in 2D space. As shown in Figure 3, we present an example. Previous sampling methods (Luo et al., 2024b) often require a long path to find $x_0$,

and reducing NFE can lead to cumulative errors, making it impossible to locate $\boldsymbol{x}_0$. In contrast, our algorithm produces more direct trajectories, allowing us to find $\boldsymbol{x}_0$ with fewer NFEs.

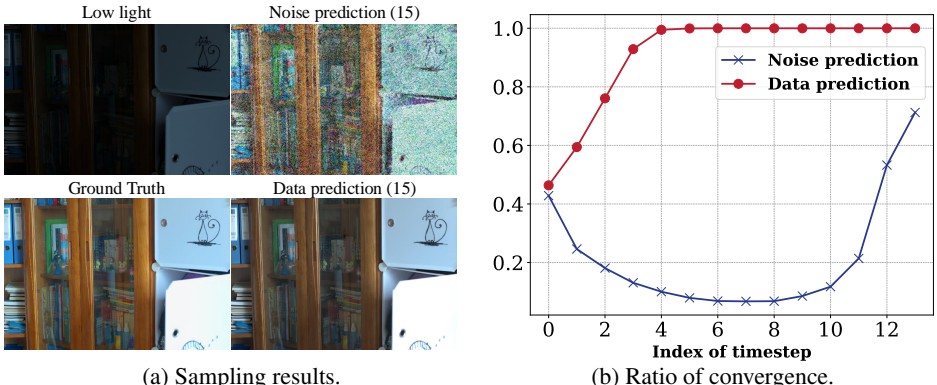

(a) Sampling results.

(b) Ratio of convergence.

Figure 4: **Convergence of noise prediction and data prediction.** In (a), we choose a low-light image for example. The numbers in parentheses indicate the NFE. In (b), we illustrate the ratio of components of neural network output that satisfy the Taylor expansion convergence requirement.

**Numerical stability of parameterizations.** From Table 1, we observe poor sampling results for noise prediction in the case of few NFEs. The reason may be that the neural network parameterized by noise prediction is numerically unstable. Recall that we used Taylor expansion in Eq.(14), and the condition for the equality to hold is $|\lambda - \lambda_s| < \boldsymbol{R}(s)$. And the radius of convergence $\boldsymbol{R}(t)$ can be calculated by

$$\frac{1}{\boldsymbol{R}(t)} = \lim_{n \to \infty} \left| \frac{\boldsymbol{c}_{n+1}(t)}{\boldsymbol{c}_n(t)} \right|, \tag{30}$$

where $\boldsymbol{c}_n(t)$ is the coefficient of the $n$-th term in Taylor expansion. We are unable to compute this limit and can only compute the $n = 0$ case as an approximation. The output of the neural network can be viewed as a vector, with each component corresponding to a radius of convergence. At each time step, we count the ratio of components that satisfy $\boldsymbol{R}_i(s) > |\lambda - \lambda_s|$ as a criterion for judging the convergence, where $i$ denotes the $i$-th component. As shown in Figure 4, the neural network parameterized by data prediction meets the convergence criteria at almost every step. However, the neural network parameterized by noise prediction always has components that cannot converge, which will lead to large errors and failed sampling. Therefore, data prediction has better numerical stability and is a more recommended choice.

## 6 CONCLUSION

We have developed a the fast sampling algorithm of MR Diffusion. Compared with DPMs, MR Diffusion is different in SDE and thus not adaptable to existing training-free fast samplers. We propose MR Sampler for acceleration of sampling of MR Diffusion. We solve the reverse-time SDE and PF-ODE derived from MRSDE and find a semi-analytical solution. We adopt the methods of *exponential integrators* to estimate the non-linear integral part. Abundant experiments demonstrate that our algorithm achieves small errors and fast convergence. Additionally, we visualize sampling trajectories and explain why the parameterization of noise prediction does not perform well in the case of small NFEs.

**Limitations and broader impact.** Despite the effectiveness of MR Sampler, our method is still inferior to distillation methods (Song et al., 2023; Luo et al., 2023a) within less than 5 NFEs. Additionally, our method can only accelerate sampling, but cannot improve the upper limit of sampling quality.

## REPRODUCIBILITY STATEMENT

Our codes are based on the official code of MR Diffusion (Luo et al., 2023b) and DPM-Solver (Lu et al., 2022b). And we use the checkpoints and datasets provided by MR Diffusion (Luo et al., 2023b). We will release them after the blind review.

## ACKNOWLEDGEMENT

This work was supported in part by the National Natural Science Foundation of China (grant 92354307), the National Key Research and Development Program of China (grant 2024YFF0729202), the Strategic Priority Research Program of the Chinese Academy of Sciences (grant XDA0460305), and the Fundamental Research Funds for the Central Universities (grant E3E45201X2). This work was also supported by Alibaba Group through Alibaba Research Intern Program.

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

## APPENDIX

We include several appendices with derivations, additional details and results. In Appendix A, we provide derivations of propositions in Section 3 and 4, equivalence between *posterior sampling* and Euler-Maruyama discretization, and velocity prediction, respectively. In Appendix B, we compare the notations used in this paper and MRSDE (Luo et al., 2023b). In Appendix C, we list detailed algorithms of MR Sampler with various orders and parameterizations. In Appendix D, we present details about datasets, settings and results in experiments. In Appendix E, we provide an in-depth discussion on determining the optimal NFE.

## A  DERIVATION DETAILS

### A.1  PROOFS OF PROPOSITIONS

**Proposition 1.** Given an initial value $\boldsymbol{x}_s$ at time $s \in [0, T]$, the solution $\boldsymbol{x}_t$ at time $t \in [0, s]$ of Eq.(10) is

$$\boldsymbol{x}_t = \frac{\alpha_t}{\alpha_s}\boldsymbol{x}_s + (1 - \frac{\alpha_t}{\alpha_s})\boldsymbol{\mu} + \alpha_t \int_s^t g^2(\tau)\frac{\boldsymbol{\epsilon}_\theta(\boldsymbol{x}_\tau, \tau)}{\alpha_\tau\sigma_\tau}\mathrm{d}\tau + \sqrt{-\int_s^t \frac{\alpha_t^2}{\alpha_\tau^2}g^2(\tau)\mathrm{d}\tau}\boldsymbol{z}, \qquad (31)$$

where $\alpha_t := e^{-\int_0^t f(\tau)\mathrm{d}\tau}$ and $\boldsymbol{z} \sim \mathcal{N}(\boldsymbol{0}, \boldsymbol{I})$.

*Proof.* For SDEs in the form of Eq.(1), Itô's formula gives the following conclusion:

$$\mathrm{d}\psi(\boldsymbol{x}, t) = \frac{\partial\psi(\boldsymbol{x}, t)}{\partial t}\mathrm{d}t + \frac{\partial\psi(\boldsymbol{x}, t)}{\partial\boldsymbol{x}}[f(\boldsymbol{x}, t)\mathrm{d}t + g(t)\mathrm{d}w] + \frac{1}{2}\frac{\partial^2\psi(\boldsymbol{x}, t)}{\partial\boldsymbol{x}^2}g^2(t)\mathrm{d}t, \qquad (32)$$

where $\psi(\boldsymbol{x}, t)$ is a differentiable function. And we define

$$\psi(\boldsymbol{x}, t) = \boldsymbol{x}e^{\int_0^t f(\tau)\mathrm{d}\tau}$$

By substituting $f(\boldsymbol{x}, t)$ and $g(t)$ with the corresponding drift and diffusion coefficients in Eq.(10), we obtain

$$\mathrm{d}\psi(\boldsymbol{x}, t) = \boldsymbol{\mu}f(t)e^{\int_0^t f(\tau)\mathrm{d}\tau}\mathrm{d}t + e^{\int_0^t f(\tau)\mathrm{d}\tau}\left[\frac{g^2(t)}{\sigma_t}\boldsymbol{\epsilon}_\theta(\boldsymbol{x}_t, t)\mathrm{d}t + g(t)\mathrm{d}\bar{w}\right].$$

And we integrate both sides of the above equation from $s$ to $t$:

$$\psi(\boldsymbol{x}, t) - \psi(\boldsymbol{x}, s) = \boldsymbol{\mu}(e^{\int_0^t f(\tau)\mathrm{d}\tau} - e^{\int_0^s f(\tau)\mathrm{d}\tau}) + \int_s^t e^{\int_0^\tau f(\xi)\mathrm{d}\xi}g^2(\tau)\frac{\boldsymbol{\epsilon}_\theta(\boldsymbol{x}_\tau, \tau)}{\sigma_\tau}\mathrm{d}\tau + \int_s^t e^{\int_0^\tau f(\xi)\mathrm{d}\xi}g(\tau)\mathrm{d}\bar{w}.$$

Note that $\bar{w}$ is a standard Wiener process running backwards in time and we have the quadratic variation $(\mathrm{d}\bar{w})^2 = -\mathrm{d}\tau$. According to the definition of $\psi(\boldsymbol{x}, t)$ and $\alpha_t$, we have

$$\frac{\boldsymbol{x}_t}{\alpha_t} - \frac{\boldsymbol{x}_s}{\alpha_s} = \boldsymbol{\mu}\left(\frac{1}{\alpha_t} - \frac{1}{\alpha_s}\right) + \int_s^t g^2(\tau)\frac{\boldsymbol{\epsilon}_\theta(\boldsymbol{x}_\tau, \tau)}{\alpha_\tau\sigma_\tau}\mathrm{d}\tau + \sqrt{-\int_s^t \frac{g^2(\tau)}{\alpha_\tau^2}\mathrm{d}\tau}\boldsymbol{z},$$

which is equivalent to Eq.(31).

**Proposition 2.** Given an initial value $\boldsymbol{x}_s$ at time $s \in [0, T]$, the solution $\boldsymbol{x}_t$ at time $t \in [0, s]$ of Eq.(15) is

$$\boldsymbol{x}_t = \frac{\alpha_t}{\alpha_s}\boldsymbol{x}_s + (1 - \frac{\alpha_t}{\alpha_s})\boldsymbol{\mu} + \alpha_t \int_s^t \frac{g^2(\tau)}{2\alpha_\tau\sigma_\tau}\boldsymbol{\epsilon}_\theta(\boldsymbol{x}_\tau, \tau)\mathrm{d}\tau, \qquad (33)$$

where $\alpha_t := e^{-\int_0^t f(\tau)\mathrm{d}\tau}$.

*Proof.* For ODEs which have a semi-linear structure as follows:

$$\frac{\mathrm{d}\boldsymbol{x}}{\mathrm{d}t} = P(t)\boldsymbol{x} + Q(\boldsymbol{x}, t), \qquad (34)$$

the method of "variation of constants" gives the following solution:

$$\boldsymbol{x}(t) = e^{\int_0^t P(\tau)\mathrm{d}\tau} \cdot \left[\int_0^t Q(\boldsymbol{x}, \tau)e^{-\int_0^\tau P(r)\mathrm{d}r}\mathrm{d}\tau + C\right].$$

By simultaneously considering the following two equations

$$
\begin{cases}
\boldsymbol{x}(t) = e^{\int_0^t P(\tau)\mathrm{d}\tau} \cdot \left[ \int_0^t Q(\boldsymbol{x}, \tau) e^{-\int_0^\tau P(r)\mathrm{d}r} \mathrm{d}\tau + C \right], \\
\boldsymbol{x}(s) = e^{\int_0^s P(\tau)\mathrm{d}\tau} \cdot \left[ \int_0^s Q(\boldsymbol{x}, \tau) e^{-\int_0^\tau P(r)\mathrm{d}r} \mathrm{d}\tau + C \right],
\end{cases}
$$

and eliminating $C$, we obtain

$$
\boldsymbol{x}(t) = \boldsymbol{x}(s) e^{\int_s^t P(\tau)\mathrm{d}\tau} + \int_s^t Q(\boldsymbol{x}, \tau) e^{\int_\tau^t P(\xi)\mathrm{d}\xi} \mathrm{d}\tau. \tag{35}
$$

Now we compare Eq.(15) with Eq.(34) and let

$$
P(t) = -f(t)
$$

$$
\text{and } Q(\boldsymbol{x}, t) = f(t)\boldsymbol{\mu} + \frac{g^2(t)}{2\sigma_t} \boldsymbol{\epsilon}_\theta(\boldsymbol{x}_t, t).
$$

Therefore, we can rewrite Eq.(35) as

$$
\begin{aligned}
\boldsymbol{x}_t &= \boldsymbol{x}_s e^{-\int_s^t f(\tau)\mathrm{d}\tau} + \int_s^t e^{-\int_\tau^t f(\xi)\mathrm{d}\xi} \left[ f(\tau)\boldsymbol{\mu} + \frac{g^2(\tau)}{2\sigma_\tau} \boldsymbol{\epsilon}_\theta(\boldsymbol{x}_\tau, \tau) \right] \mathrm{d}\tau \\
&= \boldsymbol{x}_s e^{-\int_s^t f(\tau)\mathrm{d}\tau} + \boldsymbol{\mu}(1 - e^{-\int_s^t f(\tau)\mathrm{d}\tau}) + \int_s^t e^{-\int_\tau^t f(\xi)\mathrm{d}\xi} \frac{g^2(\tau)}{2\sigma_\tau} \boldsymbol{\epsilon}_\theta(\boldsymbol{x}_\tau, \tau) \mathrm{d}\tau,
\end{aligned}
$$

which is equivalent to Eq.(33).

**Proposition 3.** Given an initial value $\boldsymbol{x}_s$ at time $s \in [0, T]$, the solution $\boldsymbol{x}_t$ at time $t \in [0, s]$ of Eq.(22) is

$$
\begin{aligned}
\boldsymbol{x}_t = {} & \frac{\sigma_t}{\sigma_s} e^{-(\lambda_t - \lambda_s)} \boldsymbol{x}_s + \boldsymbol{\mu} \left( 1 - \frac{\alpha_t}{\alpha_s} e^{-2(\lambda_t - \lambda_s)} - \alpha_t + \alpha_t e^{-2(\lambda_t - \lambda_s)} \right) \\
& + 2\alpha_t \int_{\lambda_s}^{\lambda_t} e^{-2(\lambda_t - \lambda)} \boldsymbol{x}_\theta(\boldsymbol{x}_\lambda, \lambda) \mathrm{d}\lambda + \sigma_t \sqrt{1 - e^{-2(\lambda_t - \lambda_s)}} \boldsymbol{z},
\end{aligned} \tag{36}
$$

where $\boldsymbol{z} \sim \mathcal{N}(\boldsymbol{0}, \boldsymbol{I})$.

*Proof.* According to Eq.(32), we define

$$
u(t) = \frac{g^2(t)}{\sigma_t^2} - f(t)
$$

$$
\text{and} \quad \psi(\boldsymbol{x}, t) = \boldsymbol{x} e^{\int_0^t u(\tau)\mathrm{d}\tau}.
$$

We substitute $f(\boldsymbol{x}, t)$ and $g(t)$ in Eq.(32) with the corresponding drift and diffusion coefficients in Eq.(22), and integrate both sides of the equation from $s$ to $t$:

$$
\begin{aligned}
\boldsymbol{x}_t = {} & \boldsymbol{x}_s e^{\int_s^t u(\tau)\mathrm{d}\tau} + \boldsymbol{\mu} \int_s^t e^{\int_\tau^t u(\xi)\mathrm{d}\xi} \left[ f(\tau) - \frac{g^2(\tau)}{\sigma_\tau^2}(1 - \alpha_\tau) \right] \mathrm{d}\tau \\
& - \int_s^t e^{\int_\tau^t u(\xi)\mathrm{d}\xi} \left[ \frac{g^2(\tau)}{\sigma_\tau^2} \alpha_\tau \boldsymbol{x}_\theta(\boldsymbol{x}_\tau, \tau) \right] \mathrm{d}\tau + \int_s^t e^{\int_\tau^t u(\xi)\mathrm{d}\xi} g(\tau) \mathrm{d}\bar{\boldsymbol{w}}.
\end{aligned} \tag{37}
$$

We can rewrite $g(\tau)$ as Eq.(12) and obtain

$$
e^{\int_s^t u(\tau)\mathrm{d}\tau} = \exp \int_s^t \left( -2\frac{\mathrm{d}\lambda_\tau}{\mathrm{d}\tau} - f(\tau) \right) \mathrm{d}\tau = \frac{\alpha_t}{\alpha_s} e^{-2(\lambda_t - \lambda_s)} = \frac{\sigma_t}{\sigma_s} e^{-(\lambda_t - \lambda_s)}. \tag{38}
$$

Next, we consider each term in Eq.(37) by employing Eq.(12) and Eq.(38). Firstly, we simplify the second term:

$$\boldsymbol{\mu} \int_s^t e^{\int_\tau^t u(\xi)\mathrm{d}\xi} \left[ f(\tau) - \frac{g^2(\tau)}{\sigma_\tau^2}(1 - \alpha_\tau) \right] \mathrm{d}\tau$$

$$= \boldsymbol{\mu} \int_s^t \frac{\sigma_t}{\sigma_\tau} e^{-(\lambda_t - \lambda_\tau)} \left[ f(\tau) + 2(1 - \alpha_\tau)\frac{\mathrm{d}\lambda_\tau}{\mathrm{d}\tau} \right] \mathrm{d}\tau$$

$$= \boldsymbol{\mu}\sigma_t e^{-\lambda_t} \int_s^t \frac{e^{\lambda_\tau}}{\sigma_\tau} \left[ f(\tau)\mathrm{d}\tau + 2(1 - \alpha_\tau)\mathrm{d}\lambda_\tau \right]$$

$$= \boldsymbol{\mu}\sigma_t e^{-\lambda_t} \int_s^t \frac{\alpha_\tau}{\sigma_\tau^2} \left[ f(\tau)\mathrm{d}\tau + 2\mathrm{d}\lambda_\tau - 2\alpha_\tau\mathrm{d}\lambda_\tau \right]$$

$$= \boldsymbol{\mu}\sigma_t e^{-\lambda_t} \int_s^t \frac{-\mathrm{d}\alpha_\tau}{\sigma_\tau^2} + \frac{2\alpha_\tau}{\sigma_\tau^2}\mathrm{d}\lambda_\tau - 2\frac{\alpha_\tau^2}{\sigma_\tau^2}\mathrm{d}\lambda_\tau. \tag{39}$$

Note that

$$\mathrm{d}\lambda_t = \mathrm{d}\left( \log \frac{\alpha_t}{\sigma_\infty \sqrt{1 - \alpha_t^2}} \right) = \frac{\mathrm{d}\alpha_t}{\alpha_t} + \frac{\alpha_t \mathrm{d}\alpha_t}{1 - \alpha_t^2} = \frac{\mathrm{d}\alpha_t}{\alpha_t(1 - \alpha_t^2)}. \tag{40}$$

Substitute Eq.(40) into Eq.(39) and we obtain

$$\boldsymbol{\mu} \int_s^t e^{\int_\tau^t u(\xi)\mathrm{d}\xi} \left[ f(\tau) - \frac{g^2(\tau)}{\sigma_\tau^2}(1 - \alpha_\tau) \right] \mathrm{d}\tau$$

$$= \boldsymbol{\mu}\sigma_t e^{-\lambda_t} \int_s^t \frac{-\mathrm{d}\alpha_\tau}{\sigma_\tau^2} + \frac{2\mathrm{d}\alpha_\tau}{\sigma_\tau^2(1 - \alpha_\tau^2)} - 2\frac{\alpha_\tau^2}{\sigma_\tau^2}\mathrm{d}\lambda_\tau$$

$$= \boldsymbol{\mu}\sigma_t e^{-\lambda_t} \int_s^t \frac{1 + \alpha_\tau^2}{\sigma_\infty^2(1 - \alpha_\tau^2)^2}\mathrm{d}\alpha_\tau - 2e^{2\lambda_\tau}\mathrm{d}\lambda_\tau$$

$$= \boldsymbol{\mu}\sigma_t e^{-\lambda_t} \int_s^t \frac{1}{\sigma_\infty^2}\mathrm{d}\left( \frac{\alpha_\tau}{1 - \alpha_\tau^2} \right) - 2e^{2\lambda_\tau}\mathrm{d}\lambda$$

$$= \boldsymbol{\mu}\left( 1 - \frac{\alpha_t}{\alpha_s}e^{-2(\lambda_t - \lambda_s)} - \alpha_t + \alpha_t e^{-2(\lambda_t - \lambda_s)} \right). \tag{41}$$

Secondly, we rewrite the third term in Eq.(37) by employing Eq.(12) and Eq.(38).

$$-\int_s^t e^{\int_\tau^t u(\xi)\mathrm{d}\xi} \left[ \frac{g^2(\tau)}{\sigma_\tau^2}\alpha_\tau \boldsymbol{x}_\theta(\boldsymbol{x}_\tau, \tau) \right] \mathrm{d}\tau = -\int_s^t \frac{\sigma_t}{\sigma_\tau} e^{-(\lambda_t - \lambda_\tau)} \left[ -2\frac{\mathrm{d}\lambda_\tau}{\mathrm{d}\tau}\alpha_\tau \boldsymbol{x}_\theta(\boldsymbol{x}_\tau, \tau) \right] \mathrm{d}\tau$$

$$= 2\int_s^t \sigma_t e^{2\lambda_\tau - \lambda_t} \boldsymbol{x}_\theta(\boldsymbol{x}_\tau, \lambda_\tau)\mathrm{d}\lambda_\tau$$

$$= 2\alpha_t \int_{\lambda_s}^{\lambda_t} e^{-2(\lambda_t - \lambda)} \boldsymbol{x}_\theta(\boldsymbol{x}_\lambda, \lambda)\mathrm{d}\lambda. \tag{42}$$

Thirdly, we consider the fourth term in Eq.(37) (note that $(\mathrm{d}\bar{\boldsymbol{w}})^2 = -\mathrm{d}\tau$):

$$\int_s^t e^{\int_\tau^t u(\xi)\mathrm{d}\xi} g(\tau)\mathrm{d}\bar{\boldsymbol{w}} = \sqrt{-\int_s^t e^{2\int_\tau^t u(\xi)\mathrm{d}\xi} g^2(\tau)\mathrm{d}\tau}\,\boldsymbol{z}$$

$$= \sqrt{-\int_s^t \frac{\sigma_t^2}{\sigma_\tau^2} e^{-2(\lambda_t - \lambda_\tau)} \left( -2\sigma_\tau^2 \frac{\mathrm{d}\lambda_\tau}{\mathrm{d}\tau} \right) \mathrm{d}\tau}\,\boldsymbol{z}$$

$$= \sqrt{\sigma_t^2 \int_s^t 2e^{2(\lambda_\tau - \lambda_t)}\mathrm{d}\lambda_\tau}\,\boldsymbol{z}$$

$$= \sigma_t \sqrt{1 - e^{-2(\lambda_t - \lambda_s)}}\,\boldsymbol{z}. \tag{43}$$

Lastly, we substitute Eq.(38) and Eq.(41-43) into Eq.(37) and obtain the solution as presented in Eq.(36).

**Proposition 4.** Given an initial value $\boldsymbol{x}_s$ at time $s \in [0, T]$, the solution $\boldsymbol{x}_t$ at time $t \in [0, s]$ of Eq.(24) is

$$\boldsymbol{x}_t = \frac{\sigma_t}{\sigma_s}\boldsymbol{x}_s + \boldsymbol{\mu}\left(1 - \frac{\sigma_t}{\sigma_s} + \frac{\sigma_t}{\sigma_s}\alpha_s - \alpha_t\right) + \sigma_t \int_{\lambda_s}^{\lambda_t} e^\lambda \boldsymbol{x}_\theta(\boldsymbol{x}_\lambda, \lambda)\mathrm{d}\lambda. \tag{44}$$

*Proof.* Note that Eq.(24) shares the same structure as Eq.(34). Let

$$P(t) = \frac{g^2(t)}{2\sigma_t^2} - f(t),$$

$$\text{and} \quad Q(\boldsymbol{x}, t) = \left[f(t) - \frac{g^2(t)}{2\sigma_t^2}(1 - \alpha_t)\right]\boldsymbol{\mu} - \frac{g^2(t)}{2\sigma_t^2}\alpha_t\boldsymbol{x}_\theta(\boldsymbol{x}_t, t).$$

According to Eq.(12), we first consider

$$e^{\int_s^t P(\tau)\mathrm{d}\tau} = \exp\int_s^t\left[\frac{g^2(\tau)}{2\sigma_\tau^2} - f(\tau)\right]\mathrm{d}\tau = \exp\int_s^t -\mathrm{d}\lambda_\tau + \mathrm{d}\log\alpha_\tau$$

$$= \exp\int_s^t \mathrm{d}\log\alpha_\tau - \mathrm{d}\log\frac{\alpha_\tau}{\sigma_\tau} = \exp\int_s^t \mathrm{d}\log\sigma_\tau = \frac{\sigma_t}{\sigma_s}. \tag{45}$$

Then, we can rewrite Eq.(35) as

$$\boldsymbol{x}_t = \frac{\sigma_t}{\sigma_s}\boldsymbol{x}_s + \boldsymbol{\mu}\int_s^t \frac{\sigma_t}{\sigma_\tau}\left[f(\tau) - \frac{g^2(\tau)}{2\sigma_\tau^2}(1 - \alpha_\tau)\right]\mathrm{d}\tau - \int_s^t \frac{\sigma_t}{\sigma_\tau}\frac{g^2(\tau)}{2\sigma_\tau^2}\alpha_\tau\boldsymbol{x}_\theta(\boldsymbol{x}_\tau, \tau)\mathrm{d}\tau. \tag{46}$$

Firstly, we consider the second term in Eq.(46)

$$\boldsymbol{\mu}\int_s^t \frac{\sigma_t}{\sigma_\tau}\left[f(\tau) - \frac{g^2(\tau)}{2\sigma_\tau^2}(1 - \alpha_\tau)\right]\mathrm{d}\tau$$

$$= \boldsymbol{\mu}\sigma_t\int_s^t \frac{1}{\sigma_\tau}\left[f(\tau) - \frac{g^2(\tau)}{2\sigma_\tau^2} + \frac{g^2(\tau)}{2\sigma_\tau^2}\alpha_\tau\right]\mathrm{d}\tau$$

$$= \boldsymbol{\mu}\sigma_t\left[\int_s^t \frac{1}{\sigma_\tau}\left(f(\tau) - \frac{g^2(\tau)}{2\sigma_\tau^2}\right)\mathrm{d}\tau + \int_s^t \frac{g^2(\tau)}{2\sigma_\tau^3}\alpha_\tau\mathrm{d}\tau\right]$$

$$= \boldsymbol{\mu}\sigma_t\left[-\int_s^t \frac{\mathrm{d}\log\sigma_\tau}{\sigma_\tau} - \int_s^t \frac{\alpha_\tau}{\sigma_\tau}\mathrm{d}\lambda_\tau\right] \quad \text{(refer to Eq.(12) and Eq.(45))}$$

$$= \boldsymbol{\mu}\sigma_t\left[\int_s^t \mathrm{d}\left(\frac{1}{\sigma_\tau}\right) - \int_s^t \mathrm{d}e^{\lambda_\tau}\right]$$

$$= \boldsymbol{\mu}\left(1 - \frac{\sigma_t}{\sigma_s} + \frac{\sigma_t}{\sigma_s}\alpha_s - \alpha_t\right). \tag{47}$$

Secondly, we rewrite the third term in Eq.(46)

$$-\int_s^t \frac{\sigma_t}{\sigma_\tau}\frac{g^2(\tau)}{2\sigma_\tau^2}\alpha_\tau\boldsymbol{x}_\theta(\boldsymbol{x}_\tau, \tau)\mathrm{d}\tau = \sigma_t\int_s^t e^{\lambda_\tau}\boldsymbol{x}_\theta(\boldsymbol{x}_\lambda, \lambda)\mathrm{d}\lambda_\tau. \tag{48}$$

By substituting Eq.(47) and Eq.(48) into Eq.(46), we can obtain the solution shown in Eq.(44).

## A.2 Equivalence between Posterior Sampling and Euler-Maruyama Discretization

The *posterior sampling* (Luo et al., 2024b) algorithm utilizes the reparameterization of Gaussian distribution in Eq.(19) and computes $\boldsymbol{x}_{i-1}$ from $\boldsymbol{x}_i$ iteratively as follows:

$$\boldsymbol{x}_{i-1} = \tilde{\boldsymbol{\mu}}_i(\boldsymbol{x}_i, \boldsymbol{x}_0) + \sqrt{\tilde{\beta}_i}\boldsymbol{z}_i,$$

$$\tilde{\boldsymbol{\mu}}_i(\boldsymbol{x}_i, \boldsymbol{x}_0) = \frac{(1 - \alpha_{i-1}^2)\alpha_i}{(1 - \alpha_i^2)\alpha_{i-1}}(\boldsymbol{x}_i - \boldsymbol{\mu}) + \frac{1 - \frac{\alpha_i^2}{\alpha_{i-1}^2}}{1 - \alpha_i^2}\alpha_{i-1}(\boldsymbol{x}_0 - \boldsymbol{\mu}) + \boldsymbol{\mu}, \tag{49}$$

$$\tilde{\beta}_i = \frac{(1 - \alpha_{i-1}^2)(1 - \frac{\alpha_i^2}{\alpha_{i-1}^2})}{1 - \alpha_i^2},$$

where $z_i \sim \mathcal{N}(\mathbf{0}, I)$, $\alpha_i = e^{-\int_0^i f(\tau)\mathrm{d}\tau}$ and $\boldsymbol{x}_0 = (\boldsymbol{x}_i - \boldsymbol{\mu} - \sigma_i \boldsymbol{\epsilon}_\theta(\boldsymbol{x}_i, \boldsymbol{\mu}, t_i))/\alpha_i + \boldsymbol{\mu}$. By substituting $\boldsymbol{x}_0$ into $\tilde{\boldsymbol{\mu}}_i$, we arrange the equation and obtain

$$
\begin{aligned}
\tilde{\boldsymbol{\mu}}_i(\boldsymbol{x}_i, \boldsymbol{x}_0) &= \frac{\alpha_{i-1}}{\alpha_i}\boldsymbol{x}_i + (1 - \frac{\alpha_{i-1}}{\alpha_i})\boldsymbol{\mu} - \frac{\frac{\alpha_{i-1}}{\alpha_i} - \frac{\alpha_i}{\alpha_{i-1}}}{1 - \alpha_i^2}\sigma_i \tilde{\boldsymbol{\epsilon}}_\theta(\boldsymbol{x}_i, \boldsymbol{\mu}, t_i) \\
&= \frac{\alpha_{i-1}}{\alpha_i}\boldsymbol{x}_i + (1 - \frac{\alpha_{i-1}}{\alpha_i})\boldsymbol{\mu} - \frac{\frac{\alpha_{i-1}}{\alpha_i} - \frac{\alpha_i}{\alpha_{i-1}}}{\sqrt{1 - \alpha_i^2}}\sigma_\infty \tilde{\boldsymbol{\epsilon}}_\theta(\boldsymbol{x}_i, \boldsymbol{\mu}, t_i).
\end{aligned}
\tag{50}
$$

We note that

$$
\frac{\alpha_{i-1}}{\alpha_i} = e^{\int_{i-1}^i f(\tau)\mathrm{d}\tau} = 1 + \int_{i-1}^i f(\tau)\mathrm{d}\tau + o\left(\int_{i-1}^i f(\tau)\mathrm{d}\tau\right) \approx 1 + f(t_i)\Delta t_i,
\tag{51}
$$

where the high-order error term is omitted and $\Delta t_i := t_i - t_{i-1}$. By substituting Eq.(51) into Eq.(50) and Eq.(49), we obtain

$$
\tilde{\boldsymbol{\mu}}_i(\boldsymbol{x}_i, \boldsymbol{x}_0) = (1 + f(t_i)\Delta t_i)\boldsymbol{x}_i - f(t_i)\Delta t_i \boldsymbol{\mu} - \frac{2f(t_i)\Delta t_i \sigma_\infty}{\sqrt{1 - \alpha_i^2}}\tilde{\boldsymbol{\epsilon}}_\theta(\boldsymbol{x}_i, \boldsymbol{\mu}, t_i),
\tag{52}
$$

$$
\tilde{\beta}_i = \frac{(1 - \alpha_{i-1}^2)(1 - \frac{\alpha_i^2}{\alpha_{i-1}^2})}{1 - \alpha_i^2} \approx \frac{2f(t_i)\Delta t_i(1 - \alpha_{i-1}^2)}{1 - \alpha_i^2}.
\tag{53}
$$

On the other hand, the reverse-time SDE has been presented in Eq.(10). Combining the assumption $g^2(t)/f(t) = 2\sigma_\infty^2$ in Section 2.2 and the definition of $\sigma_t$ in Section 3.1, the Euler–Maruyama descretization of this SDE is

$$
\boldsymbol{x}_{i-1} - \boldsymbol{x}_i = -f(t_i)(\boldsymbol{\mu} - \boldsymbol{x}_i)\Delta t_i - \frac{g^2(t_i)}{\sigma_i}\tilde{\boldsymbol{\epsilon}}_\theta(\boldsymbol{x}_i, \boldsymbol{\mu}, t_i)\Delta t_i + g(t_i)\sqrt{\Delta t_i}\boldsymbol{z}_i,
$$

$$
\begin{aligned}
\therefore \boldsymbol{x}_{i-1} &= (1 + f(t_i)\Delta t_i)\boldsymbol{x}_i - f(t_i)\Delta t_i \boldsymbol{\mu} - \frac{2\sigma_\infty^2 f(t_i)}{\sigma_\infty\sqrt{1 - \alpha_i^2}}\tilde{\boldsymbol{\epsilon}}_\theta(\boldsymbol{x}_i, \boldsymbol{\mu}, t_i)\Delta t_i + g(t_i)\sqrt{\Delta t_i}\boldsymbol{z}_i \\
&= (1 + f(t_i)\Delta t_i)\boldsymbol{x}_i - f(t_i)\Delta t_i \boldsymbol{\mu} - \frac{2f(t_i)\Delta t_i \sigma_\infty}{\sqrt{1 - \alpha_i^2}}\tilde{\boldsymbol{\epsilon}}_\theta(\boldsymbol{x}_i, \boldsymbol{\mu}, t_i) + \sigma_\infty\sqrt{2f(t_i)\Delta t_i}\boldsymbol{z}_i \\
&= \tilde{\boldsymbol{\mu}}_i(\boldsymbol{x}_i, \boldsymbol{x}_0) + \sigma_\infty\sqrt{\frac{1 - \alpha_i^2}{1 - \alpha_{i-1}^2}\tilde{\beta}_i}\boldsymbol{z}_i.
\end{aligned}
\tag{54}
$$

Thus, the *posterior sampling* algorithm is a special Euler–Maruyama descretization of reverse-time SDE with a different coefficient of Gaussian noise.

## A.3 DERIVATIONS ABOUT VELOCITY PREDICTION

Following Eq.(27), We can define the *velocity prediction* as

$$
\boldsymbol{v}_\theta(t) = \boldsymbol{\mu}\sin\phi_t - \boldsymbol{x}_\theta(t)\sin\phi_t + \sigma_\infty\cos(\phi_t)\boldsymbol{\epsilon}_\theta(t).
\tag{55}
$$

And we have the relationship between $\boldsymbol{x}_\theta(t)$ and $\boldsymbol{\epsilon}_\theta(t)$ as follows:

$$
\boldsymbol{x}_t = \boldsymbol{x}_\theta(t)\cos\phi_t + \boldsymbol{\mu}(1 - \cos\phi_t) + \sigma_\infty\sin(\phi_t)\boldsymbol{\epsilon}_\theta(t).
\tag{56}
$$

In order to get $\boldsymbol{x}_\theta$ from $\boldsymbol{v}_\theta$, we rewrite Eq.(55) as

$$
\boldsymbol{x}_\theta(t)\sin^2\phi_t = \boldsymbol{\mu}\sin^2\phi_t - \boldsymbol{v}_\theta(t)\sin\phi_t + \sigma_\infty\boldsymbol{\epsilon}_\theta(t)\sin\phi_t\cos\phi_t.
\tag{57}
$$

Then we replace $\boldsymbol{\epsilon}_\theta(t)$ according to Eq.(56)

$$
\begin{aligned}
\boldsymbol{x}_\theta(t)\sin^2\phi_t &= \boldsymbol{\mu}\sin^2\phi_t - \boldsymbol{v}_\theta(t)\sin\phi_t + [\boldsymbol{x}_t - \boldsymbol{x}_\theta(t)\cos\phi_t - \boldsymbol{\mu}(1 - \cos\phi_t)]\cos\phi_t \\
&= (1 - \cos\phi_t)\boldsymbol{\mu} - \boldsymbol{v}_\theta(t)\sin\phi_t + \boldsymbol{x}_t\cos\phi_t - \boldsymbol{x}_\theta(t)\cos^2\phi_t.
\end{aligned}
\tag{58}
$$

Arranging the above equation, we can obtain the transformation from $\boldsymbol{v}_\theta$ to $\boldsymbol{x}_\theta$, as shown in Eq.(28). Similarly, we can also rewrite Eq.(55) and replace $\boldsymbol{x}_\theta(t)$ as follows:

$$
\begin{aligned}
\sigma_\infty\cos^2(\phi_t)\boldsymbol{\epsilon}_\theta(t) &= \boldsymbol{v}_\theta(t)\cos\phi_t - \boldsymbol{\mu}\sin\phi_t\cos\phi_t + \boldsymbol{x}_\theta(t)\sin\phi_t\cos\phi_t \\
&= \boldsymbol{v}_\theta(t)\cos\phi_t - \boldsymbol{\mu}\sin\phi_t\cos\phi_t + \sin\phi_t[\boldsymbol{x}_t - \boldsymbol{\mu}(1 - \cos\phi_t) - \sigma_\infty\sin(\phi_t)\boldsymbol{\epsilon}_\theta(t)] \\
&= \boldsymbol{v}_\theta(t)\cos\phi_t - \boldsymbol{\mu}\sin\phi_t + \boldsymbol{x}_\theta(t)\sin\phi_t - \sigma_\infty\sin^2\phi_t\boldsymbol{\epsilon}_\theta(t).
\end{aligned}
\tag{59}
$$

Thus we obtain the transformation from $\boldsymbol{v}_\theta$ to $\boldsymbol{\epsilon}_\theta$, as presented in Eq.(29).

# B  NOTATION COMPARISON TABLE

| This paper | $f(t)$ | $g(t)$ | $\alpha_t$ | $\sigma_t$ |
|---|---|---|---|---|
| MRSDE (Luo et al., 2023b) | $\theta_t$ | $\sigma_t$ | $e^{-\int_0^t \theta_\tau \mathrm{d}\tau}$ | $\sigma_\infty \sqrt{1 - e^{-2\int_0^t \theta_\tau \mathrm{d}\tau}}$ |

Table 4: The correspondence between the notations used in this paper (left column) and notations used by MRSDE (right column).

# C  DETAILED SAMPLING ALGORITHM OF MR SAMPLER

We list the detailed MR Sampler algorithm with different solvers, parameterizations and orders as follows.

---

**Algorithm 1** MR Sampler-SDE-n-1.

---

**Require:** initial value $\boldsymbol{x}_T = \boldsymbol{\mu} + \sigma_\infty \boldsymbol{\epsilon}$, Gaussian noise sequence $\{\boldsymbol{z}_i | \boldsymbol{z}_i \sim \mathcal{N}(\boldsymbol{0}, \boldsymbol{I})\}_{i=1}^M$, time steps $\{t_i\}_{i=0}^M$, data prediction model $\boldsymbol{x}_\theta$. Denote $h_i := \lambda_{t_i} - \lambda_{t_{i-1}}$ for $i = 1, \ldots, M$.

1: $\boldsymbol{x}_{t_0} \leftarrow \boldsymbol{x}_T$. Initialize an empty buffer $Q$.
2: $Q \xleftarrow{\text{buffer}} \boldsymbol{x}_\theta(\boldsymbol{x}_{t_0}, t_0)$
3: **for** $i \leftarrow 1$ to $M$ **do**
4:     $\boldsymbol{x}_{t_i} = \frac{\alpha_{t_i}}{\alpha_{t_{i-1}}} \boldsymbol{x}_{t_{i-1}} + \left(1 - \frac{\alpha_{t_i}}{\alpha_{t_{i-1}}}\right) \boldsymbol{\mu} - 2\sigma_{t_i}(e^{h_i} - 1)\boldsymbol{\epsilon}_\theta(\boldsymbol{x}_{t_{i-1}}, t_{i-1}) + \sigma_{t_i}\sqrt{e^{2h_i} - 1}\boldsymbol{z}_i$
5:     If $i < M$, then $Q \xleftarrow{\text{buffer}} \boldsymbol{x}_\theta(\boldsymbol{x}_{t_i}, t_i)$
6: **end for**
7: **return** $\boldsymbol{x}_{t_M}$

---

**Algorithm 2** MR Sampler-SDE-n-2.

---

**Require:** initial value $\boldsymbol{x}_T = \boldsymbol{\mu} + \sigma_\infty \boldsymbol{\epsilon}$, Gaussian noise sequence $\{\boldsymbol{z}_i | \boldsymbol{z}_i \sim \mathcal{N}(\boldsymbol{0}, \boldsymbol{I})\}_{i=1}^M$, time steps $\{t_i\}_{i=0}^M$, data prediction model $\boldsymbol{x}_\theta$. Denote $h_i := \lambda_{t_i} - \lambda_{t_{i-1}}$ for $i = 1, \ldots, M$.

1: $\boldsymbol{x}_{t_0} \leftarrow \boldsymbol{x}_T$. Initialize an empty buffer $Q$.
2: $Q \xleftarrow{\text{buffer}} \boldsymbol{x}_\theta(\boldsymbol{x}_{t_0}, t_0)$
3: $\boldsymbol{x}_{t_1} = \frac{\alpha_{t_1}}{\alpha_{t_0}} \boldsymbol{x}_{t_0} + \left(1 - \frac{\alpha_{t_1}}{\alpha_{t_0}}\right) \boldsymbol{\mu} - 2\sigma_{t_1}(e^{h_1} - 1)\boldsymbol{\epsilon}_\theta(\boldsymbol{x}_{t_0}, t_0) + \sigma_{t_1}\sqrt{e^{2h_1} - 1}\boldsymbol{z}_1$
4: $Q \xleftarrow{\text{buffer}} \boldsymbol{x}_\theta(\boldsymbol{x}_{t_1}, t_1)$
5: **for** $i \leftarrow 2$ to $M$ **do**
6:     $\boldsymbol{D}_i = \frac{\boldsymbol{\epsilon}_\theta(\boldsymbol{x}_{t_{i-1}}, t_{i-1}) - \boldsymbol{\epsilon}_\theta(\boldsymbol{x}_{t_{i-2}}, t_{i-2})}{h_{i-1}}$
7:     $\boldsymbol{x}_{t_i} = \frac{\alpha_{t_i}}{\alpha_{t_{i-1}}} \boldsymbol{x}_{t_{i-1}} + \left(1 - \frac{\alpha_{t_i}}{\alpha_{t_{i-1}}}\right) \boldsymbol{\mu} - 2\sigma_{t_i}\left[(e^{h_i} - 1)\boldsymbol{\epsilon}_\theta(\boldsymbol{x}_{t_{i-1}}, t_{i-1}) + (e^{h_i} - 1 - h_i)\boldsymbol{D}_i\right] + \sigma_{t_i}\sqrt{e^{2h_i} - 1}\boldsymbol{z}_i$
8:     If $i < M$, then $Q \xleftarrow{\text{buffer}} \boldsymbol{x}_\theta(\boldsymbol{x}_{t_i}, t_i)$
9: **end for**
10: **return** $\boldsymbol{x}_{t_M}$

---

---

**Algorithm 3** MR Sampler-ODE-n-1.

---

**Require:** initial value $\boldsymbol{x}_T = \boldsymbol{\mu} + \sigma_\infty \boldsymbol{\epsilon}$, time steps $\{t_i\}_{i=0}^M$, data prediction model $\boldsymbol{x}_\theta$. Denote $h_i := \lambda_{t_i} - \lambda_{t_{i-1}}$ for $i = 1, \ldots, M$.
1: $\boldsymbol{x}_{t_0} \leftarrow \boldsymbol{x}_T$. Initialize an empty buffer $Q$.
2: $Q \xleftarrow{\text{buffer}} \boldsymbol{x}_\theta(\boldsymbol{x}_{t_0}, t_0)$
3: **for** $i \leftarrow 1$ to $M$ **do**
4: $\quad \boldsymbol{x}_{t_i} = \frac{\alpha_{t_i}}{\alpha_{t_{i-1}}} \boldsymbol{x}_{t_{i-1}} + \left(1 - \frac{\alpha_{t_i}}{\alpha_{t_{i-1}}}\right) \boldsymbol{\mu} - \sigma_{t_i}(e^{h_i} - 1)\boldsymbol{\epsilon}_\theta(\boldsymbol{x}_{t_{i-1}}, t_{i-1})$
5: $\quad$ If $i < M$, then $Q \xleftarrow{\text{buffer}} \boldsymbol{x}_\theta(\boldsymbol{x}_{t_i}, t_i)$
6: **end for**
7: **return** $\boldsymbol{x}_{t_M}$

---

**Algorithm 4** MR Sampler-ODE-n-2.

---

**Require:** initial value $\boldsymbol{x}_T = \boldsymbol{\mu} + \sigma_\infty \boldsymbol{\epsilon}$, time steps $\{t_i\}_{i=0}^M$, data prediction model $\boldsymbol{x}_\theta$. Denote $h_i := \lambda_{t_i} - \lambda_{t_{i-1}}$ for $i = 1, \ldots, M$.
1: $\boldsymbol{x}_{t_0} \leftarrow \boldsymbol{x}_T$. Initialize an empty buffer $Q$.
2: $Q \xleftarrow{\text{buffer}} \boldsymbol{x}_\theta(\boldsymbol{x}_{t_0}, t_0)$
3: $\boldsymbol{x}_{t_1} = \frac{\alpha_{t_1}}{\alpha_{t_0}} \boldsymbol{x}_{t_0} + \left(1 - \frac{\alpha_{t_1}}{\alpha_{t_0}}\right) \boldsymbol{\mu} - \sigma_{t_1}(e^{h_1} - 1)\boldsymbol{\epsilon}_\theta(\boldsymbol{x}_{t_0}, t_0)$
4: $Q \xleftarrow{\text{buffer}} \boldsymbol{x}_\theta(\boldsymbol{x}_{t_1}, t_1)$
5: **for** $i \leftarrow 2$ to $M$ **do**
6: $\quad \boldsymbol{D}_i = \frac{\boldsymbol{\epsilon}_\theta(\boldsymbol{x}_{t_{i-1}}, t_{i-1}) - \boldsymbol{\epsilon}_\theta(\boldsymbol{x}_{t_{i-2}}, t_{i-2})}{h_{i-1}}$
7: $\quad \boldsymbol{x}_{t_i} = \frac{\alpha_{t_i}}{\alpha_{t_{i-1}}} \boldsymbol{x}_{t_{i-1}} + \left(1 - \frac{\alpha_{t_i}}{\alpha_{t_{i-1}}}\right) \boldsymbol{\mu} - \sigma_{t_i}\left[(e^{h_i} - 1)\boldsymbol{\epsilon}_\theta(\boldsymbol{x}_{t_{i-1}}, t_{i-1}) + (e^{h_i} - 1 - h_i)\boldsymbol{D}_i\right]$
8: $\quad$ If $i < M$, then $Q \xleftarrow{\text{buffer}} \boldsymbol{x}_\theta(\boldsymbol{x}_{t_i}, t_i)$
9: **end for**
10: **return** $\boldsymbol{x}_{t_M}$

---

**Algorithm 5** MR Sampler-SDE-d-1.

---

**Require:** initial value $\boldsymbol{x}_T = \boldsymbol{\mu} + \sigma_\infty \boldsymbol{\epsilon}$, Gaussian noise sequence $\{\boldsymbol{z}_i | \boldsymbol{z}_i \sim \mathcal{N}(\boldsymbol{0}, \boldsymbol{I})\}_{i=1}^M$, time steps $\{t_i\}_{i=0}^M$, data prediction model $\boldsymbol{x}_\theta$. Denote $h_i := \lambda_{t_i} - \lambda_{t_{i-1}}$ for $i = 1, \ldots, M$.
1: $\boldsymbol{x}_{t_0} \leftarrow \boldsymbol{x}_T$. Initialize an empty buffer $Q$.
2: $Q \xleftarrow{\text{buffer}} \boldsymbol{x}_\theta(\boldsymbol{x}_{t_0}, t_0)$
3: **for** $i \leftarrow 1$ to $M$ **do**
4: $\quad \boldsymbol{x}_{t_i} = \frac{\sigma_{t_i}}{\sigma_{t_{i-1}}} e^{-h_i} \boldsymbol{x}_{t_{i-1}} + \boldsymbol{\mu}\left(1 - \frac{\alpha_{t_i}}{\alpha_{t_{i-1}}} e^{-2h_i} - \alpha_{t_i} + \alpha_{t_i} e^{-2h_i}\right) + \sigma_{t_i}\sqrt{1 - e^{-2h_i}}\boldsymbol{z}_i + $
$\quad\quad \alpha_{t_i}\left(1 - e^{-2h_i}\right) \boldsymbol{x}_\theta(\boldsymbol{x}_{t_{i-1}}, t_{i-1})$
5: $\quad$ If $i < M$, then $Q \xleftarrow{\text{buffer}} \boldsymbol{x}_\theta(\boldsymbol{x}_{t_i}, t_i)$
6: **end for**
7: **return** $\boldsymbol{x}_{t_M}$

---

---

**Algorithm 6** MR Sampler-SDE-d-2.

---

**Require:** initial value $\boldsymbol{x}_T = \boldsymbol{\mu} + \sigma_\infty \boldsymbol{\epsilon}$, Gaussian noise sequence $\{\boldsymbol{z}_i | \boldsymbol{z}_i \sim \mathcal{N}(\boldsymbol{0}, \boldsymbol{I})\}_{i=1}^M$, time steps $\{t_i\}_{i=0}^M$, data prediction model $\boldsymbol{x}_\theta$. Denote $h_i := \lambda_{t_i} - \lambda_{t_{i-1}}$ for $i = 1, \ldots, M$.

1: $\boldsymbol{x}_{t_0} \leftarrow \boldsymbol{x}_T$. Initialize an empty buffer $Q$.

2: $Q \xleftarrow{\text{buffer}} \boldsymbol{x}_\theta(\boldsymbol{x}_{t_0}, t_0)$

3: $\boldsymbol{x}_{t_1} = \frac{\sigma_{t_1}}{\sigma_{t_0}} e^{-h_1} \boldsymbol{x}_{t_0} + \boldsymbol{\mu} \left(1 - \frac{\alpha_{t_1}}{\alpha_{t_0}} e^{-2h_1} - \alpha_{t_1} + \alpha_{t_1} e^{-2h_1}\right) + \alpha_{t_1} \left(1 - e^{-2h_1}\right) \boldsymbol{x}_\theta(\boldsymbol{x}_{t_0}, t_0) + \sigma_{t_1} \sqrt{1 - e^{-2h_1}} \boldsymbol{z}_1$

4: $Q \xleftarrow{\text{buffer}} \boldsymbol{x}_\theta(\boldsymbol{x}_{t_i}, t_i)$

5: **for** $i \leftarrow 2$ to $M$ **do**

6:      $\boldsymbol{D}_i = \frac{\boldsymbol{x}_\theta(\boldsymbol{x}_{t_{i-1}}, t_{i-1}) - \boldsymbol{x}_\theta(\boldsymbol{x}_{t_{i-2}}, t_{i-2})}{h_{i-1}}$

7:      $\boldsymbol{x}_{t_i} = \frac{\sigma_{t_i}}{\sigma_{t_{i-1}}} e^{-h_i} \boldsymbol{x}_{t_{i-1}} + \boldsymbol{\mu} \left(1 - \frac{\alpha_{t_i}}{\alpha_{t_{i-1}}} e^{-2h_i} - \alpha_{t_i} + \alpha_{t_i} e^{-2h_i}\right) + \sigma_{t_i} \sqrt{1 - e^{-2h_i}} \boldsymbol{z}_i + \alpha_{t_i} \left(1 - e^{-2h_i}\right) \boldsymbol{x}_\theta(\boldsymbol{x}_{t_{i-1}}, t_{i-1}) + \alpha_{t_i} \left(h_i - \frac{1 - e^{-2h_i}}{2}\right) \boldsymbol{D}_i$

8:      If $i < M$, then $Q \xleftarrow{\text{buffer}} \boldsymbol{x}_\theta(\boldsymbol{x}_{t_i}, t_i)$

9: **end for**

10: **return** $\boldsymbol{x}_{t_M}$

---

**Algorithm 7** MR Sampler-ODE-d-1.

---

**Require:** initial value $\boldsymbol{x}_T = \boldsymbol{\mu} + \sigma_\infty \boldsymbol{\epsilon}$, time steps $\{t_i\}_{i=0}^M$, data prediction model $\boldsymbol{x}_\theta$. Denote $h_i := \lambda_{t_i} - \lambda_{t_{i-1}}$ for $i = 1, \ldots, M$.

1: $\boldsymbol{x}_{t_0} \leftarrow \boldsymbol{x}_T$. Initialize an empty buffer $Q$.

2: $Q \xleftarrow{\text{buffer}} \boldsymbol{x}_\theta(\boldsymbol{x}_{t_0}, t_0)$

3: **for** $i \leftarrow 1$ to $M$ **do**

4:      $\boldsymbol{x}_{t_i} = \frac{\sigma_{t_i}}{\sigma_{t_{i-1}}} \boldsymbol{x}_{t_{i-1}} + \boldsymbol{\mu} \left(1 - \frac{\sigma_{t_i}}{\sigma_{t_{i-1}}} + \frac{\sigma_{t_i}}{\sigma_{t_{i-1}}} \alpha_{t_{i-1}} - \alpha_{t_i}\right) + \alpha_{t_i} \left(1 - e^{-h_i}\right) \boldsymbol{x}_\theta(\boldsymbol{x}_{t_{i-1}}, t_{i-1})$

5:      If $i < M$, then $Q \xleftarrow{\text{buffer}} \boldsymbol{x}_\theta(\boldsymbol{x}_{t_i}, t_i)$

6: **end for**

7: **return** $\boldsymbol{x}_{t_M}$

---

**Algorithm 8** MR Sampler-ODE-d-2.

---

**Require:** initial value $\boldsymbol{x}_T = \boldsymbol{\mu} + \sigma_\infty \boldsymbol{\epsilon}$, time steps $\{t_i\}_{i=0}^M$, data prediction model $\boldsymbol{x}_\theta$. Denote $h_i := \lambda_{t_i} - \lambda_{t_{i-1}}$ for $i = 1, \ldots, M$.

1: $\boldsymbol{x}_{t_0} \leftarrow \boldsymbol{x}_T$. Initialize an empty buffer $Q$.

2: $Q \xleftarrow{\text{buffer}} \boldsymbol{x}_\theta(\boldsymbol{x}_{t_0}, t_0)$

3: $\boldsymbol{x}_{t_1} = \frac{\sigma_{t_1}}{\sigma_{t_0}} \boldsymbol{x}_{t_0} + \boldsymbol{\mu} \left(1 - \frac{\sigma_{t_1}}{\sigma_{t_0}} + \frac{\sigma_{t_1}}{\sigma_{t_0}} \alpha_{t_0} - \alpha_{t_1}\right) + \alpha_{t_1} \left(1 - e^{-h_1}\right) \boldsymbol{x}_\theta(\boldsymbol{x}_{t_0}, t_0)$

4: $Q \xleftarrow{\text{buffer}} \boldsymbol{x}_\theta(\boldsymbol{x}_{t_1}, t_1)$

5: **for** $i \leftarrow 2$ to $M$ **do**

6:      $\boldsymbol{x}_{t_i} = \frac{\sigma_{t_i}}{\sigma_{t_{i-1}}} \boldsymbol{x}_{t_{i-1}} + \boldsymbol{\mu} \left(1 - \frac{\sigma_{t_i}}{\sigma_{t_{i-1}}} + \frac{\sigma_{t_i}}{\sigma_{t_{i-1}}} \alpha_{t_{i-1}} - \alpha_{t_i}\right) + \alpha_{t_i} \left(1 - e^{-h_i}\right) \boldsymbol{x}_\theta(\boldsymbol{x}_{t_{i-1}}, t_{i-1}) + \alpha_{t_i} \left(h_i - 1 + e^{-h_i}\right) \frac{\boldsymbol{x}_\theta(\boldsymbol{x}_{t_{i-1}}, t_{i-1}) - \boldsymbol{x}_\theta(\boldsymbol{x}_{t_{i-2}}, t_{i-2})}{h_{i-1}}$

7:      If $i < M$, then $Q \xleftarrow{\text{buffer}} \boldsymbol{x}_\theta(\boldsymbol{x}_{t_i}, t_i)$

8: **end for**

9: **return** $\boldsymbol{x}_{t_M}$

---

# D DETAILS ABOUT EXPERIMENTS

## D.1 DETAILS ABOUT DATASETS

We list details about the used datesets in 10 image restoration tasks in Table 5.

| Task name | Dataset name | Reference | Number of testing images |
|---|---|---|---|
| Blurry | GoPro | Nah et al. (2017) | 1111 |
| Hazy | RESIDE-6k | Qin et al. (2020a) | 1000 |
| JPEG-compressing | DIV2K, Flickr2K and LIVE1 | Agustsson & Timofte (2017),Timofte et al. (2017), Sheikh (2005) | 29 |
| Low-light | LOL | Wei et al. (2018) | 15 |
| Noisy | DIV2K, Flickr2K and CBSD68 | Agustsson & Timofte (2017),Timofte et al. (2017),Martin et al. (2001) | 68 |
| Raindrop | RainDrop | Qian et al. (2018) | 58 |
| Rainy | Rain100H | (Yang et al., 2017) | 100 |
| Shadowed | SRD | (Qu et al., 2017) | 408 |
| Snowy | Snow100K-L | (Liu et al., 2018) | 601 |
| Inpainting | CelebaHQ | (Lugmayr et al., 2022) | 100 |

Table 5: **Details about the used datasets in 10 image restoration tasks**

## D.2 DETAILS ON THE NEURAL NETWORK ARCHITECTURE

In this section, we describe the neural network architecture used in experiments. We follow the framework of Luo et al. (2024a), an image restoration model designed to address multiple degradation problems simultaneously without requiring prior knowledge of the degradation. The diffusion model in Luo et al. (2024a) is derived from Luo et al. (2023c), and its neural network architecture is based on *NAFNet*. NAFNet builds upon the U-Net architecture by replacing traditional activation functions with *SimpleGate* and incorporating an additional multi-layer perceptron to manage channel scaling and offset parameters for embedding temporal information into the attention and feedforward layers. For further details, please refer to Section 4.2 in Luo et al. (2023c).

## D.3 DETAILED METRICS ON ALL TASKS

We list results on four metrics for ten image restoration tasks in Table 6-15.

| NFE | Method | LPIPS↓ | FID↓ | PSNR↑ | SSIM↑ |
|---|---|---|---|---|---|
| 100 | Posterior | **0.0385** | 18.35 | **29.49** | **0.9102** |
| | Euler | 0.0426 | 20.71 | 29.34 | 0.8981 |
| | MR Sampler-1 | 0.0390 | **18.16** | 29.45 | 0.9086 |
| | MR Sampler-2 | 0.0397 | 18.81 | 29.30 | 0.9055 |
| 50 | Posterior | 0.4238 | 247.0 | 12.85 | 0.6048 |
| | Euler | 0.4449 | 249.5 | 12.68 | 0.5800 |
| | MR Sampler-1 | **0.0379** | **18.03** | **29.68** | **0.9118** |
| | MR Sampler-2 | 0.0402 | 19.09 | 29.15 | 0.9046 |
| 20 | Posterior | 0.7130 | 347.4 | 10.19 | 0.2171 |
| | Euler | 0.7257 | 344.3 | 10.16 | 0.2073 |
| | MR Sampler-1 | **0.0383** | 18.29 | **30.05** | **0.9172** |
| | MR Sampler-2 | 0.0408 | 19.13 | 28.98 | 0.9032 |
| 10 | Posterior | 0.8097 | 374.1 | 9.802 | 0.1339 |
| | Euler | 0.8154 | 381.1 | 9.786 | 0.1305 |
| | MR Sampler-1 | **0.0401** | 18.46 | **30.61** | **0.9229** |
| | MR Sampler-2 | 0.0437 | 19.29 | 28.48 | 0.8996 |
| 5 | Posterior | 0.8489 | 385.6 | 9.599 | 0.1057 |
| | Euler | 0.8525 | 384.7 | 9.587 | 0.1042 |
| | MR Sampler-1 | **0.0428** | **20.00** | **31.03** | **0.9262** |
| | MR Sampler-2 | 0.0529 | 24.02 | 28.35 | 0.8930 |

Table 6: **Image inpainting.**

| NFE | Method | LPIPS↓ | FID↓ | PSNR↑ | SSIM↑ |
|---|---|---|---|---|---|
| 100 | Posterior | 0.0614 | 21.42 | 27.43 | **0.8763** |
| | Euler | 0.0683 | 23.27 | 27.09 | 0.8577 |
| | MR Sampler-1 | **0.0608** | **21.30** | **27.45** | 0.8754 |
| | MR Sampler-2 | 0.0626 | 21.47 | 27.18 | 0.8691 |
| 50 | Posterior | 0.2374 | 72.04 | 21.35 | 0.7037 |
| | Euler | 0.2730 | 76.02 | 21.02 | 0.6676 |
| | MR Sampler-1 | **0.0602** | **20.91** | **27.63** | **0.8803** |
| | MR Sampler-2 | 0.0628 | 21.85 | 27.08 | 0.8685 |
| 20 | Posterior | 0.6622 | 123.8 | 16.42 | 0.3546 |
| | Euler | 0.6861 | 126.0 | 16.23 | 0.3431 |
| | MR Sampler-1 | **0.0601** | 21.32 | **28.07** | **0.8903** |
| | MR Sampler-2 | 0.0650 | 22.34 | 26.89 | 0.8645 |
| 10 | Posterior | 0.8013 | 138.5 | 14.76 | 0.2694 |
| | Euler | 0.8164 | 140.4 | 14.64 | 0.2640 |
| | MR Sampler-1 | **0.0608** | 22.26 | **28.50** | **0.8992** |
| | MR Sampler-2 | 0.0698 | 23.92 | 26.49 | 0.8573 |
| 5 | Posterior | 0.8590 | 145.8 | 13.92 | 0.2318 |
| | Euler | 0.8680 | 145.8 | 13.85 | 0.2290 |
| | MR Sampler-1 | **0.0611** | 23.29 | **28.89** | **0.9065** |
| | MR Sampler-2 | 0.0628 | **21.95** | 27.06 | 0.8718 |

Table 7: **Snowy image restoration.**

| NFE | Method | LPIPS↓ | FID↓ | PSNR↑ | SSIM↑ |
|---|---|---|---|---|---|
| 100 | Posterior | **0.0970** | 20.14 | **27.84** | **0.8391** |
| | Euler | 0.1129 | 20.30 | 27.59 | 0.8112 |
| | MR Sampler-1 | 0.0984 | **20.03** | 27.73 | 0.8370 |
| | MR Sampler-2 | 0.0989 | 20.69 | 27.44 | 0.8329 |
| 50 | Posterior | 0.6119 | 101.8 | 18.29 | 0.4720 |
| | Euler | 0.6985 | 114.2 | 17.80 | 0.4123 |
| | MR Sampler-1 | **0.0978** | 20.90 | **27.92** | **0.8409** |
| | MR Sampler-2 | 0.1006 | **20.74** | 27.20 | 0.8310 |
| 20 | Posterior | 1.043 | 187.2 | 14.73 | 0.2049 |
| | Euler | 1.065 | 192.9 | 14.56 | 0.1955 |
| | MR Sampler-1 | **0.0954** | **19.79** | **28.31** | **0.8505** |
| | MR Sampler-2 | 0.1014 | 20.93 | 27.17 | 0.8299 |
| 10 | Posterior | 1.122 | 208.4 | 13.67 | 0.1525 |
| | Euler | 1.133 | 209.7 | 13.57 | 0.1484 |
| | MR Sampler-1 | **0.0956** | **19.87** | **28.67** | **0.8554** |
| | MR Sampler-2 | 0.1044 | 21.85 | 26.99 | 0.8276 |
| 5 | Posterior | 1.155 | 218.9 | 13.12 | 0.1298 |
| | Euler | 1.161 | 221.4 | 13.06 | 0.1278 |
| | MR Sampler-1 | **0.0964** | **20.16** | **28.90** | **0.8601** |
| | MR Sampler-2 | 0.2203 | 36.69 | 23.73 | 0.6690 |

Table 8: **Shadowed image restoration.**

| NFE | Method | LPIPS↓ | FID↓ | PSNR↑ | SSIM↑ |
|---|---|---|---|---|---|
| 100 | Posterior | **0.0594** | **25.58** | 29.14 | **0.8704** |
| | Euler | 0.0725 | 28.80 | 28.77 | 0.8473 |
| | MR Sampler-1 | **0.0594** | 30.53 | **29.15** | 0.8679 |
| | MR Sampler-2 | 0.0616 | 27.33 | 28.92 | 0.8614 |
| 50 | Posterior | 0.4418 | 183.1 | 16.41 | 0.4903 |
| | Euler | 0.4560 | 185.2 | 16.24 | 0.4729 |
| | MR Sampler-1 | **0.0586** | 30.73 | **29.34** | **0.8730** |
| | MR Sampler-2 | 0.0620 | **27.65** | 28.85 | 0.8602 |
| 20 | Posterior | 0.6865 | 293.6 | 12.54 | 0.2464 |
| | Euler | 0.6943 | 299.0 | 12.49 | 0.2402 |
| | MR Sampler-1 | **0.0604** | 31.19 | **29.81** | **0.8845** |
| | MR Sampler-2 | 0.0635 | **27.79** | 28.60 | 0.8559 |
| 10 | Posterior | 0.7972 | 323.0 | 11.50 | 0.1755 |
| | Euler | 0.8043 | 330.8 | 11.46 | 0.1724 |
| | MR Sampler-1 | **0.0659** | 31.66 | **30.28** | **0.8943** |
| | MR Sampler-2 | 0.0678 | **29.54** | 28.09 | 0.8483 |
| 5 | Posterior | 0.8663 | 332.4 | 10.96 | 0.1450 |
| | Euler | 0.8714 | 332.5 | 10.94 | 0.1435 |
| | MR Sampler-1 | 0.0729 | 32.06 | **30.68** | **0.9029** |
| | MR Sampler-2 | **0.0637** | **26.92** | 28.82 | 0.8685 |

Table 9: **Rainy image restoration.**

| NFE | Method | LPIPS↓ | FID↓ | PSNR↑ | SSIM↑ |
|---|---|---|---|---|---|
| 100 | Posterior | 0.0443 | **19.70** | **30.05** | **0.8910** |
| | Euler | 0.0634 | 24.19 | 29.31 | 0.8438 |
| | MR Sampler-1 | **0.0437** | 19.94 | 29.91 | 0.8852 |
| | MR Sampler-2 | 0.0454 | 21.35 | 29.55 | 0.8768 |
| 50 | Posterior | 0.4289 | 100.1 | 23.19 | 0.4663 |
| | Euler | 0.5011 | 111.6 | 22.35 | 0.4106 |
| | MR Sampler-1 | **0.0428** | 19.14 | **30.07** | **0.8914** |
| | MR Sampler-2 | 0.0459 | 20.50 | 29.40 | 0.8764 |
| 20 | Posterior | 0.8873 | 190.8 | 17.37 | 0.1925 |
| | Euler | 0.9082 | 194.1 | 17.10 | 0.1839 |
| | MR Sampler-1 | **0.0439** | **19.31** | **30.44** | **0.9025** |
| | MR Sampler-2 | 0.0470 | 21.08 | 29.34 | 0.8745 |
| 10 | Posterior | 0.9884 | 215.5 | 15.67 | 0.1419 |
| | Euler | 0.9993 | 213.1 | 15.52 | 0.1381 |
| | MR Sampler-1 | **0.0466** | **20.60** | **30.77** | **0.9114** |
| | MR Sampler-2 | 0.0485 | 22.17 | 29.37 | 0.8779 |
| 5 | Posterior | 1.030 | 226.3 | 14.82 | 0.1209 |
| | Euler | 1.037 | 226.4 | 14.74 | 0.1190 |
| | MR Sampler-1 | **0.0497** | **21.18** | **31.04** | **0.9175** |
| | MR Sampler-2 | 0.0733 | 28.26 | 28.03 | 0.8369 |

Table 10: **Raindrop image restoration.**

| NFE | Method | LPIPS↓ | FID↓ | PSNR↑ | SSIM↑ |
|---|---|---|---|---|---|
| 100 | Posterior | 0.1694 | 65.79 | **25.97** | **0.7267** |
| | Euler | 0.2719 | 68.69 | 24.28 | 0.5686 |
| | MR Sampler-1 | 0.1629 | **59.12** | **25.97** | 0.7244 |
| | MR Sampler-2 | **0.1586** | 64.02 | 25.67 | 0.7126 |
| 50 | Posterior | 0.7713 | 135.7 | 18.19 | 0.2763 |
| | Euler | 0.8060 | 143.5 | 17.74 | 0.2615 |
| | MR Sampler-1 | 0.1680 | 65.14 | **26.20** | **0.7330** |
| | MR Sampler-2 | **0.1615** | **64.24** | 25.61 | 0.7127 |
| 20 | Posterior | 0.9941 | 181.6 | 14.76 | 0.1821 |
| | Euler | 1.006 | 188.3 | 14.61 | 0.1781 |
| | MR Sampler-1 | 0.1872 | 71.31 | **26.68** | **0.7494** |
| | MR Sampler-2 | **0.1695** | **64.90** | 25.46 | 0.7098 |
| 10 | Posterior | 1.057 | 202.6 | 13.59 | 0.1550 |
| | Euler | 1.063 | 207.0 | 13.51 | 0.1529 |
| | MR Sampler-1 | 0.2043 | 79.28 | **27.13** | **0.7628** |
| | MR Sampler-2 | **0.1853** | **70.19** | 25.09 | 0.6984 |
| 5 | Posterior | 1.087 | 213.5 | 13.00 | 0.1419 |
| | Euler | 1.091 | 218.6 | 12.96 | 0.1408 |
| | MR Sampler-1 | **0.2046** | 80.45 | **27.51** | **0.7743** |
| | MR Sampler-2 | 0.3178 | **73.93** | 24.32 | 0.5485 |

Table 11: **Noisy image restoration.**

| NFE | Method | LPIPS↓ | FID↓ | PSNR↑ | SSIM↑ |
|---|---|---|---|---|---|
| 100 | Posterior | 0.0796 | 34.70 | **23.84** | **0.8496** |
| | Euler | 0.1014 | 37.46 | 23.27 | 0.8027 |
| | MR Sampler-1 | **0.0774** | 32.91 | 23.80 | 0.8451 |
| | MR Sampler-2 | 0.0789 | **32.91** | 23.59 | 0.8394 |
| 50 | Posterior | 0.6572 | 151.3 | 9.490 | 0.2746 |
| | Euler | 0.7517 | 176.7 | 9.402 | 0.2340 |
| | MR Sampler-1 | **0.0784** | 34.45 | 23.51 | **0.8476** |
| | MR Sampler-2 | 0.0786 | **32.85** | 23.52 | 0.8382 |
| 20 | Posterior | 1.288 | 390.1 | 8.211 | 0.0648 |
| | Euler | 1.297 | 396.8 | 8.212 | 0.0625 |
| | MR Sampler-1 | **0.0791** | 35.28 | **24.22** | **0.8586** |
| | MR Sampler-2 | 0.0792 | **32.80** | 23.63 | 0.8399 |
| 10 | Posterior | 1.351 | 432.2 | 8.130 | 0.0476 |
| | Euler | 1.354 | 424.2 | 8.136 | 0.0467 |
| | MR Sampler-1 | **0.0831** | 41.10 | **24.04** | **0.8619** |
| | MR Sampler-2 | 0.0841 | **36.53** | 23.22 | 0.8398 |
| 5 | Posterior | 1.371 | 453.0 | 8.114 | 0.0408 |
| | Euler | 1.372 | 447.2 | 8.118 | 0.0405 |
| | MR Sampler-1 | 0.0860 | 41.81 | 24.02 | **0.8676** |
| | MR Sampler-2 | **0.0782** | **33.98** | **24.13** | 0.8507 |

Table 12: **Low-light image restoration.**

| NFE | Method | LPIPS↓ | FID↓ | PSNR↑ | SSIM↑ |
|---|---|---|---|---|---|
| 100 | Posterior | 0.1702 | 45.77 | 25.78 | **0.7380** |
| | Euler | 0.2949 | 56.99 | 23.84 | 0.5398 |
| | MR Sampler-1 | 0.1636 | **43.67** | **25.81** | 0.7362 |
| | MR Sampler-2 | **0.1555** | 44.58 | 25.46 | 0.7220 |
| 50 | Posterior | 0.6494 | 103.7 | 19.34 | 0.2908 |
| | Euler | 0.7035 | 113.7 | 18.62 | 0.2659 |
| | MR Sampler-1 | 0.1734 | 47.09 | **26.06** | **0.7470** |
| | MR Sampler-2 | **0.1567** | **45.56** | 25.41 | 0.7224 |
| 20 | Posterior | 0.8252 | 140.6 | 16.44 | 0.1988 |
| | Euler | 0.8477 | 144.5 | 16.19 | 0.1921 |
| | MR Sampler-1 | 0.1993 | 51.43 | **26.58** | **0.7649** |
| | MR Sampler-2 | **0.1675** | **47.36** | 25.33 | 0.7235 |
| 10 | Posterior | 0.8723 | 158.9 | 15.49 | 0.1738 |
| | Euler | 0.8859 | 159.9 | 15.34 | 0.1701 |
| | MR Sampler-1 | 0.2183 | 57.83 | **26.98** | **0.7771** |
| | MR Sampler-2 | **0.1871** | **51.25** | 25.08 | 0.7197 |
| 5 | Posterior | 0.8941 | 163.6 | 14.99 | 0.1615 |
| | Euler | 0.9013 | 162.0 | 14.91 | 0.1596 |
| | MR Sampler-1 | **0.2281** | **59.17** | **27.28** | **0.7853** |
| | MR Sampler-2 | 0.3751 | 62.60 | 23.15 | 0.4797 |

Table 13: **JPEG image restoration.**

| NFE | Method | LPIPS↓ | FID↓ | PSNR↑ | SSIM↑ |
|---|---|---|---|---|---|
| 100 | Posterior | **0.0211** | **4.755** | 30.37 | **0.9485** |
| | Euler | 0.0358 | 6.182 | 29.95 | 0.9319 |
| | MR Sampler-1 | 0.0219 | 4.826 | **30.42** | 0.9462 |
| | MR Sampler-2 | 0.0230 | 4.978 | 30.26 | 0.9431 |
| 50 | Posterior | 0.3994 | 35.47 | 15.34 | 0.5808 |
| | Euler | 0.4745 | 42.70 | 15.16 | 0.5160 |
| | MR Sampler-1 | **0.0211** | **4.737** | **30.49** | **0.9484** |
| | MR Sampler-2 | 0.0233 | 4.993 | 30.19 | 0.9427 |
| 20 | Posterior | 0.9911 | 114.0 | 12.81 | 0.1832 |
| | Euler | 1.012 | 118.5 | 12.71 | 0.1752 |
| | MR Sampler-1 | **0.0200** | **4.682** | **30.63** | **0.9534** |
| | MR Sampler-2 | 0.0240 | 5.077 | 30.06 | 0.9409 |
| 10 | Posterior | 1.116 | 144.0 | 11.94 | 0.1261 |
| | Euler | 1.128 | 147.3 | 11.87 | 0.1228 |
| | MR Sampler-1 | **0.0197** | **4.785** | **30.80** | **0.9579** |
| | MR Sampler-2 | 0.0246 | 5.228 | 29.65 | 0.9372 |
| 5 | Posterior | 1.162 | 159.1 | 11.47 | 0.1042 |
| | Euler | 1.168 | 161.2 | 11.43 | 0.1026 |
| | MR Sampler-1 | **0.0205** | **4.926** | **30.56** | **0.9604** |
| | MR Sampler-2 | 0.0228 | 5.174 | 29.65 | 0.9416 |

Table 14: **Hazy image restoration.**

| NFE | Method | LPIPS↓ | FID↓ | PSNR↑ | SSIM↑ |
|---|---|---|---|---|---|
| 100 | Posterior | 0.1249 | **14.48** | **27.48** | **0.8442** |
| | Euler | 0.1404 | 16.06 | 27.16 | 0.8179 |
| | MR Sampler-1 | **0.1239** | 14.61 | 27.46 | 0.8419 |
| | MR Sampler-2 | 0.1248 | 14.61 | 27.30 | 0.8365 |
| 50 | Posterior | 0.5112 | 45.13 | 24.41 | 0.5571 |
| | Euler | 0.5739 | 61.21 | 23.79 | 0.4957 |
| | MR Sampler-1 | 0.244 | **14.47** | **27.59** | **0.8461** |
| | MR Sampler-2 | **0.1251** | 14.62 | 27.24 | 0.8360 |
| 20 | Posterior | 1.069 | 117.8 | 18.20 | 0.1717 |
| | Euler | 1.089 | 120.8 | 17.94 | 0.1637 |
| | MR Sampler-1 | 0.1287 | 14.92 | **27.85** | **0.8544** |
| | MR Sampler-2 | **0.1266** | **14.82** | 27.13 | 0.8337 |
| 10 | Posterior | 1.187 | 141.9 | 16.11 | 0.1136 |
| | Euler | 1.197 | 143.7 | 15.96 | 0.1104 |
| | MR Sampler-1 | 0.1356 | 15.65 | **28.10** | **0.8613** |
| | MR Sampler-2 | **0.1300** | **15.51** | 26.88 | 0.8295 |
| 5 | Posterior | 1.228 | 155.3 | 15.08 | 0.0922 |
| | Euler | 1.234 | 157.3 | 15.00 | 0.0907 |
| | MR Sampler-1 | 0.1422 | 16.32 | **28.31** | **0.8668** |
| | MR Sampler-2 | **0.1248** | **14.20** | 26.92 | 0.8354 |

Table 15: **Motion-blurry image restoration.**

## D.4 DETAILS ON NUMERICAL STABILITY AT LOW NFES

We have included further details regarding numerical stability at 5 NFEs to complement the experiments presented in Section 5.3. As illustrated in Figure 5, when the NFE is relatively low, the convergence rate of noise prediction at each step does not exceed 40%, which results in sampling collapse. In contrast, during the final 1 or 2 steps, the convergence rate of data prediction approaches nearly 100%.

## D.5 WALL CLOCK TIME

We measured the wall clock time in our experiments on a single NVIDIA A800 GPU. The average wall clock time per image of two representative tasks (low-light and motion-blurry image restoration) are reported in Table 16 and 17.

## D.6 PRESENTATION OF SAMPLING RESULTS

We present the sampling results for all image restoration tasks in Figure 6 and 7. We choose one image for each task.

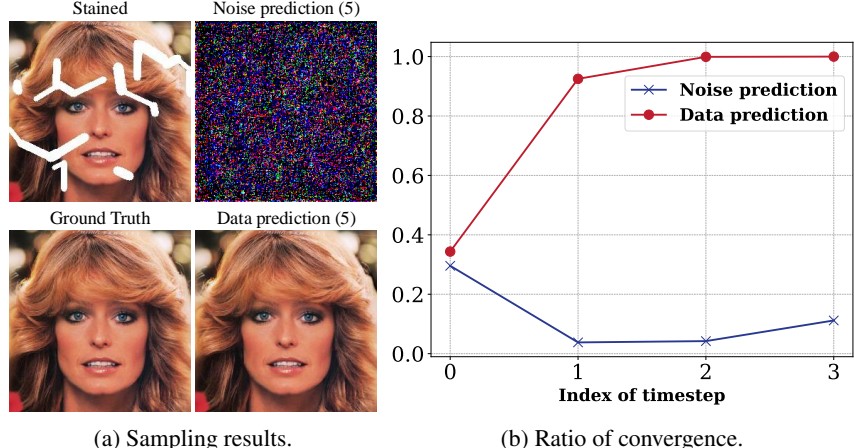

(a) Sampling results.    (b) Ratio of convergence.

Figure 5: **Convergence of noise prediction and data prediction at 5 NFEs.** In (a), we choose a stained image for example. The numbers in parentheses indicate the NFE. In (b), we illustrate the ratio of components of neural network output that satisfy the Taylor expansion convergence requirement.

Table 16: Wall clock time on the Low-light dataset.

| NFE | Method | Time(s)↓ |
|---|---|---|
| 100 | Posterior Sampling | 17.19 |
| | MR Sampler-2 | 17.83 |
| 50 | Posterior Sampling | 8.605 |
| | MR Sampler-2 | 8.439 |
| 20 | Posterior Sampling | 3.445 |
| | MR Sampler-2 | 3.285 |
| 10 | Posterior Sampling | 1.727 |
| | MR Sampler-2 | 1.569 |
| 5 | Posterior Sampling | 0.8696 |
| | MR Sampler-2 | 0.7112 |

Table 17: Wall clock time on the Motion-blurry dataset.

| NFE | Method | Time(s)↓ |
|---|---|---|
| 100 | Posterior Sampling | 82.04 |
| | MR Sampler-2 | 81.16 |
| 50 | Posterior Sampling | 41.23 |
| | MR Sampler-2 | 40.18 |
| 20 | Posterior Sampling | 16.44 |
| | MR Sampler-2 | 15.59 |
| 10 | Posterior Sampling | 8.212 |
| | MR Sampler-2 | 7.413 |
| 5 | Posterior Sampling | 4.133 |
| | MR Sampler-2 | 3.294 |

# E    SELECTION OF THE OPTIMAL NFE

In practice, sampling quality is expected to degrade as the NFE decreases, which requires us to make a trade-off between sampling quality and efficiency. From the perspective of reverse-time SDE and PF-ODE, the estimation error of the solution plays a critical role in determining the sampling quality. This estimation error is primarily influenced by $h_{max} = \max_{1 \leq i \leq M}\{\lambda_i - \lambda_{i-1}\} = \mathcal{O}(\frac{1}{M})$, where $M$ represents the NFE. During sampling, the noise schedule must be designed in advance, which also determines the $\lambda$ schedule. Since $\lambda_i$ is monotonically increasing, a larger NFE results in a smaller $h_{max}$, thereby reducing the estimation error and improving sampling quality. However, different sampling algorithms exhibit varying convergence rates. Experimental results show that the MR Sampler (our method) can achieve a good score with as few as 5 NFEs, whereas posterior sampling and Euler discretization fail to converge even with 50 NFEs.

Specifically, we conducted experiments on the snowy dataset with low NFE settings, and the results are presented in Table 18. The sampling quality remains largely stable for NFE values larger than 10, gradually deteriorates when the NFE is below 10, and collapses entirely when NFE is reduced to 2. Based on our experience, we recommend using 10–20 NFEs, which provides a reasonable trade-off between efficiency and performance.

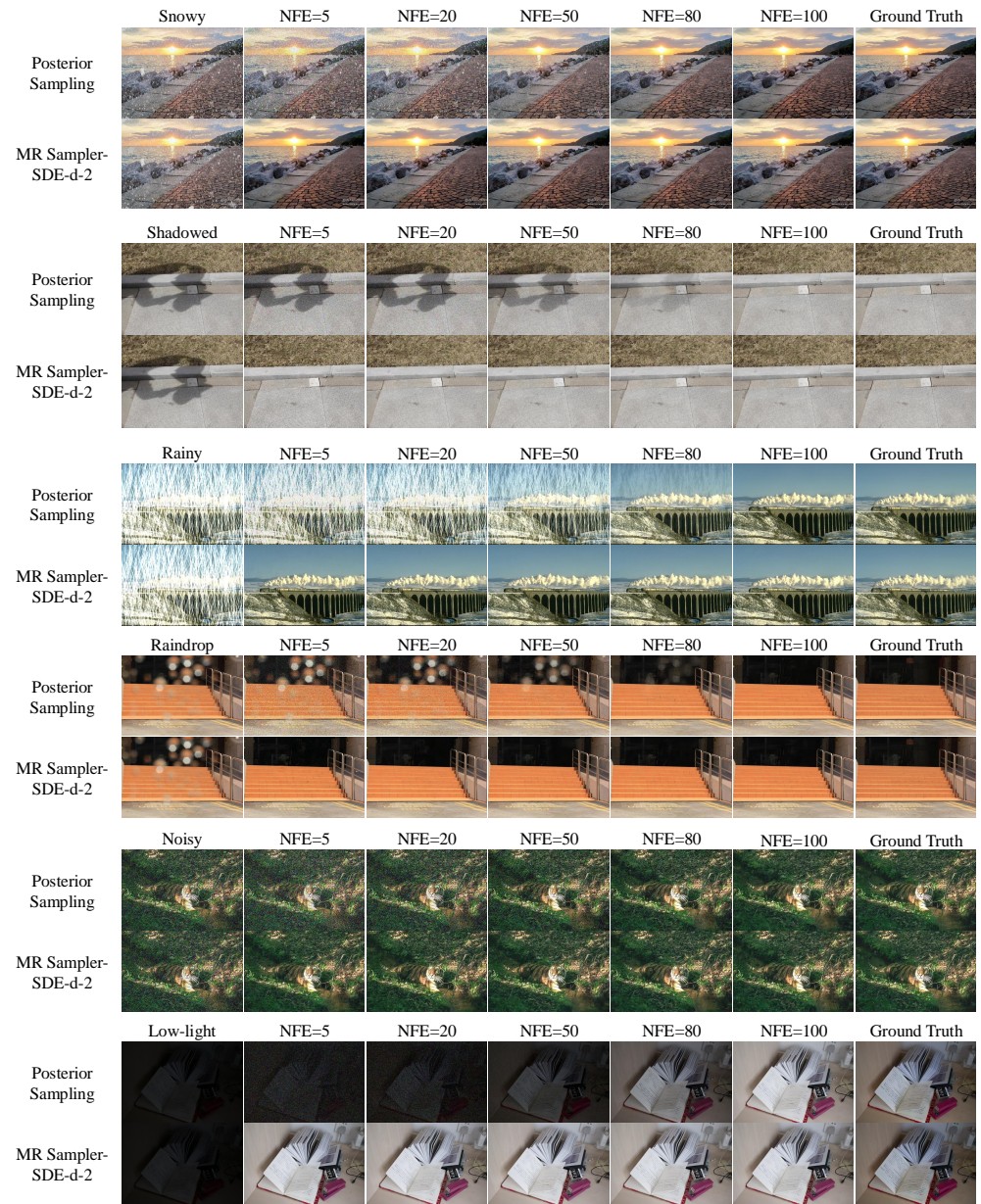

Figure 6: **Comparisons between *posterior sampling* and *MR Sampler-SDE-d-2* on snowy, shadowed, rainy, raindrop, noisy and low-light datasets.**

Table 18: Results of MR Sampler-SDE-2 with data prediction and uniform $\lambda$ on the snowy dataset.

| NFE | 100 | 50 | 20 | 10 | 8 | 6 | 5 | 4 | 2 |
|---|---|---|---|---|---|---|---|---|---|
| LPIPS↓ | 0.0626 | 0.0628 | 0.0650 | 0.0698 | 0.0725 | 0.0744 | 0.0628 | 0.1063 | 1.422 |
| FID↓ | 21.47 | 21.85 | 22.34 | 23.92 | 24.81 | 25.60 | 21.95 | 30.18 | 421.1 |
| PSNR↑ | 27.18 | 27.08 | 26.89 | 26.49 | 26.61 | 26.40 | 27.06 | 25.38 | 6.753 |
| SSIM↑ | 0.8691 | 0.8685 | 0.8645 | 0.8573 | 0.8462 | 0.8407 | 0.8718 | 0.7640 | 0.0311 |

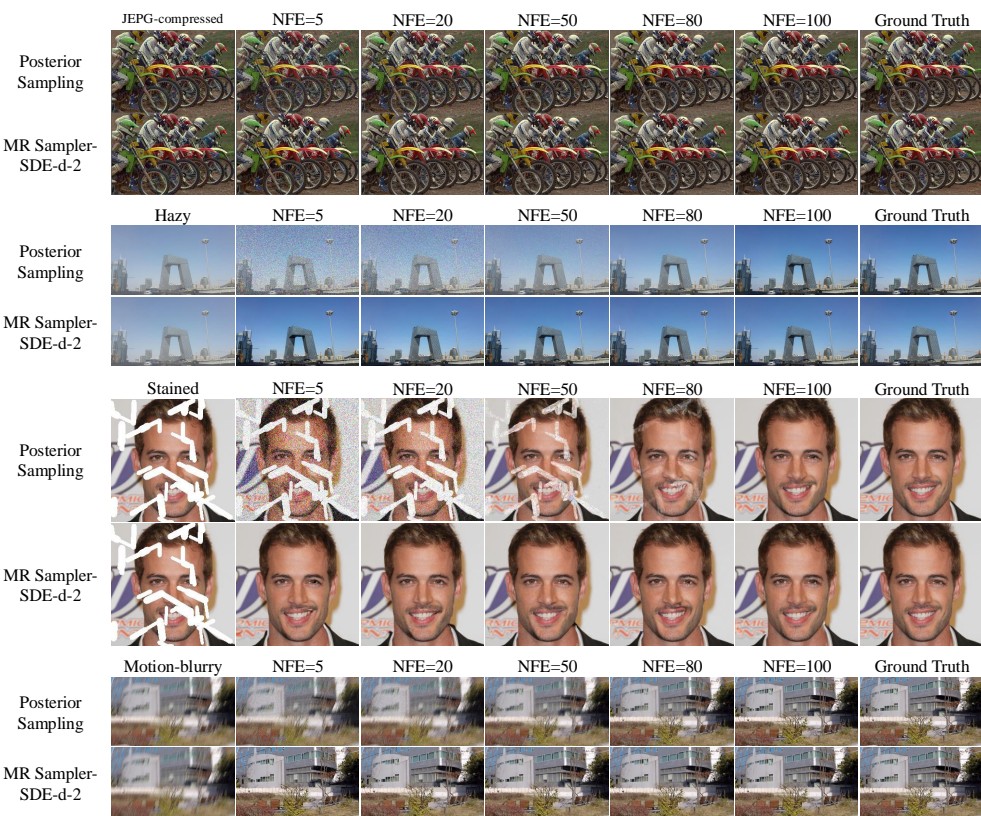

Figure 7: **Comparisons between *posterior sampling* and textitMR Sampler-SDE-d-2 on JPEG-compressed, hazy, inpainting and motion-blurry datasets.**

