# OpenReview forum: "MaRS: A Fast Sampler for Mean Reverting Diffusion based on ODE and SDE Solvers"
_ICLR.cc/2025/Conference — ICLR 2025 Spotlight_

### Official Review · Reviewer_pnRJ · 2024-10-20

**Soundness:** 3
**Presentation:** 4
**Contribution:** 3
**Rating:** 8
**Confidence:** 5

**Summary:**

The paper proposes a new perspective on improving the efficiency of sampling in Mean Reverting  Diffusion models. This approach introduces MR Sampler, which leverages both ordinary differential equations (ODE) and stochastic differential equations (SDE) to enhance the speed of the sampling process. While MR Diffusion provides a fresh perspective by modifying the structure of the SDE to make controllable image generation simpler and more natural, the main challenge it addresses is the inefficiency of sampling, which typically requires hundreds of function evaluations (NFEs) to generate high-quality outputs.

The authors build on prior work that primarily focused on denoising tasks using ODE solvers, and they extend this framework to a broader set of tasks without requiring additional training. The key technical contribution is the semi-analytical solution they derive, which reduces the number of NFEs significantly while maintaining competitive sampling quality. In fact, MR Sampler shows a speedup of 10 to 20 times across ten different image restoration tasks.

**Strengths:**

The paper introduces a novel approach to sampling in Mean Reverting  Diffusion models, which itself is a relatively new paradigm compared to more conventional diffusion processes like Variance Preserving and Variance Exploding SDEs. The key originality lies in the way the authors combine semi-analytical solutions derived from ODE and SDE solvers with MR Diffusion, enabling faster sampling. This combination is innovative as it offers a fresh perspective on accelerating diffusion models by altering the underlying stochastic differential equation structure rather than focusing solely on the score function as in prior work.

Additionally, the Mean Reverting SDE framework offers a natural integration of image conditions into the generation process, making it more applicable to tasks that require controllable generation, such as image restoration and inpainting. This is a novel contribution compared to prior work that has typically used Diffusion Schrödinger Bridge or Optimal Transport methods for similar purposes. The MR Diffusion model’s ability to address multiple tasks beyond denoising is a creative and valuable extension of existing frameworks.

The technical depth of the paper is commendable. The authors offer a rigorous derivation of the semi-analytical solution and provide substantial theoretical grounding for their approach. The use of probability flow ODEs (PF-ODEs) and reverse-time SDEs in the MR Diffusion context is clearly explained and well-justified. The semi-analytical solution, which combines an analytical function and a neural network-based integral, reduces computational complexity without compromising sampling quality.

The experiments are comprehensive and cover ten different image restoration tasks, providing strong empirical support for the proposed method. The use of multiple performance metrics (e.g., FID, LPIPS, PSNR, SSIM) demonstrates that the authors took a thorough approach to evaluating both the quality and speed of the generated samples. The speedups of 10-20x, without a significant drop in sample quality, highlight the robustness and practical value of the proposed method.

The paper is well-structured, with clear sections on the methodology, theoretical contributions, and experimental validation. The technical content, while complex, is made accessible through the use of visual aids (e.g., Figure 1 comparing qualitative sampling results, and charts showing performance metrics) and clear explanations of the mathematical formulations. The distinction between noise prediction and data prediction models, and the impact of these choices on sampling quality and stability, is clearly delineated and contributes to a deeper understanding of the technique.

The authors also provide appendices with detailed proofs and further experimental results, ensuring that the methodology is reproducible and the claims are verifiable. Overall, the clarity in presenting a technically complex subject is a strong point of the paper.

The proposed MR Sampler addresses a significant challenge in the field of diffusion models: accelerating the sampling process without sacrificing quality. The speedups achieved in this work, which range from 10x to 20x across multiple tasks, are substantial and have clear practical implications, particularly in real-time applications such as image restoration. This makes the method highly relevant for use cases that demand controllable and fast generation, such as medical imaging, video processing, and computational photography.

The method’s plug-and-play nature is another strength. It is adaptable to a variety of existing diffusion models and does not require retraining, making it easy to integrate into different applications. The broad applicability to various image restoration tasks (e.g., dehazing, inpainting, motion-blur reduction) enhances the significance of the work.

**Weaknesses:**

A notable limitation is that the paper places more emphasis on the speed of sampling rather than the quality. While the authors claim comparable quality in terms of metrics such as FID and LPIPS, they do not provide an in-depth analysis of the trade-off between sampling speed and output quality. This is a  gap, as accelerating sampling without sacrificing quality is one of the central challenges in diffusion models. The authors could have benefited from comparing their approach with other speed-quality trade-off techniques, such as those mentioned in prior work (e.g., the 'Come-Closer-Diffuse-Faster' approach). Although these methods might be older, they are relevant in establishing a clear benchmark and providing a deeper understanding of the trade-offs at play. (like https://openaccess.thecvf.com/content/CVPR2022/papers/Chung_Come-Closer-Diffuse-Faster_Accelerating_Conditional_Diffusion_Models_for_Inverse_Problems_Through_Stochastic_CVPR_2022_paper.pdf  and https://arxiv.org/abs/2108.01073 )

The experiments demonstrate a clear focus on speedups, and while they are rigorous and cover a variety of tasks (e.g., image dehazing, inpainting), the lack of comparison with more state-of-the-art methods suggests that the method may not yet be positioned as a new state of the art but rather as a faster alternative with comparable performance under specific conditions.

Overall, the introduction of the Mean Reverting SDE into the posterior sampling stage with ODE solvers is a fresh and promising perspective. However, the paper would benefit from addressing the broader implications of their method, particularly how the mean-reverting approach impacts the balance between speed and quality. Including more comprehensive comparisons with established trade-off techniques would further strengthen the contribution but overall this is a good paper, potentially worth discussing at ICLR.

**Questions:**

1. How does reducing NFEs affect sampling quality? Could you compare this with speed-quality trade-off methods like "Come-Closer-Diffuse-Faster"?

2. Do different image restoration tasks benefit differently from the MR Sampler in terms of speed or quality?

3. Why is Mean-Reverting SDE better for posterior sampling compared to other methods like Optimal Transport or Schrödinger Bridge?

4. Can you provide more details on the stability of data prediction models at low NFEs, and when would noise prediction be preferable?

5. Can MR Sampler be extended to larger or multi-modal tasks, such as video generation or text-to-image models?

6. Why are there no comparisons with SOTA models in terms of both speed and quality? Would this strengthen the paper?

---

> ### Author Response · Authors · 2024-11-22
> **Response to Reviewer pnRJ (part1)**
>
> We really appreciate the thoughtful and constructive comments of the reviewer. Below we respond to the weaknesses and questions.
>
> **Q1(a)&W1**: How does reducing NFEs affect sampling quality?
>
> **A**: From the perspective of reverse-time SDE and PF-ODE, the estimation error of the solution plays a critical role in determining the sampling quality. This estimation error is primarily influenced by $h_{max} = \max_{1 \leq i \leq M}{\lambda_i - \lambda_{i-1}} = \mathcal{O}(\frac{1}{M})$, where $M$ represents the NFE. During sampling, the noise schedule must be designed in advance, which also determines the $\lambda$ schedule. Since $\lambda_i$ is monotonically increasing, a larger NFE results in a smaller $h_{max}$, thereby reducing the estimation error and improving sampling quality.
> However, different sampling algorithms exhibit varying convergence rates. Experimental results show that the MR Sampler (our method) can achieve a good score with as few as 5 NFEs, whereas posterior sampling and Euler discretization fail to converge even with 50 NFEs.
>
>  Specifically, we conducted additional experiments on the desnow task with low NFE settings, and the results are presented in Table below. The sampling quality remains largely stable for NFE values larger than 10, gradually deteriorates when the NFE is below 10, and collapses entirely when NFE is reduced to 2.
>
> MR Sampler-SDE-2 with data prediction and uniform $\lambda$ on the desnow task
>
> | NFE  | LPIPS  |  FID  | PSNR  |  SSIM  |
> | :--: | :----: | :---: | :---: | :----: |
> | 100  | 0.0626 | 21.47 | 27.18 | 0.8691 |
> |  50  | 0.0628 | 21.85 | 27.08 | 0.8685 |
> |  20  | 0.0650 | 22.34 | 26.89 | 0.8645 |
> |  10  | 0.0698 | 23.92 | 26.49 | 0.8573 |
> |  8   | 0.0725 | 24.81 | 26.61 | 0.8462 |
> |  6   | 0.0744 | 25.60 | 26.40 | 0.8407 |
> |  5   | 0.0628 | 21.95 | 27.06 | 0.8718 |
> |  4   | 0.1063 | 30.18 | 25.38 | 0.7640 |
> |  2   | 1.422  | 421.1 | 6.753 | 0.0311 |
>
> **Q2**: Do different image restoration tasks benefit differently from the MR Sampler in terms of speed or quality?
>
> **A**: We consider that different image restoration tasks do not benefit differently from the MR Sampler in terms of speed or quality. The performance of the MR Sampler varies across different tasks; however, this variation originates from the neural network rather than the sampling algorithm itself. Our method is a sampling acceleration algorithm specifically designed for MR Diffusion and does not incorporate any task-specific prior knowledge in its derivation. From this perspective, the nature of the task does not inherently affect the performance of MR Sampler. However, our derivation involves performing the Taylor expansion on the neural network function. As a result, the performance of our algorithm is inevitably influenced by the characteristics and effectiveness of the neural network.
>
> **Q3**: Why is Mean-Reverting SDE better for posterior sampling compared to other methods like Optimal Transport or Schrödinger Bridge?
>
> **A**: Both Schrödinger Bridge and Optimal Transport approaches arrive at conclusions similar to those of MR Diffusion, albeit through different methodologies:
> $$
> q(X_t|X_0,X_1)=\mathcal{N}(X_t;\mu_t(X_0,X_1),\Sigma_t), \\quad
> \mu_t=\frac{\bar\sigma_t^2}{\bar\sigma_t^2+\sigma_t^2}X_0+\frac{\sigma_t^2}{\bar\sigma_t^2+\sigma_t^2}X_1, \\;\Sigma_t=\frac{\bar\sigma_t^2\sigma_t^2}{\bar\sigma_t^2+\sigma_t^2}I \tag{1}
> $$
>
> $$
> x_t=\mu+(x_s-\mu)e^{-\bar\theta_{s:t}}+\lambda^2(1-e^{-2\bar\theta_{s:t}})\epsilon \tag{2}
> $$
>
> Equation (1) is presented in Proposition 3.3 of paper [1], while Equation (2) is given in Proposition 3.1 of paper [2]. Despite differences in notation and definitions, the principles described by the two equations are fundamentally consistent. However, a key distinction lies in how the posterior sampling is implemented. Schrödinger Bridge adopts the posterior sampling method of DDPM without leveraging the information of $X_N$ (see Algorithm 2 in paper [1] for further details). In contrast, MR Diffusion incorporates the information of $\mu$ in its posterior sampling (refer to Equation (2) for details), which leads to improved performance.
>
> [1]. Liu, G. H., Vahdat, A., Huang, D. A., Theodorou, E. A., Nie, W., & Anandkumar, A. (2023). I $^ 2$ SB: Image-to-Image Schrödinger Bridge. *arXiv preprint arXiv:2302.05872*.
>
> [2]. Ziwei Luo, Fredrik K Gustafsson, Zheng Zhao, Jens Sjölund, and Thomas B Schön. Image restoration with mean-reverting stochastic differential equations. arXiv preprint arXiv:2301.11699, 2023b.

---

> > ### Author Response · Authors · 2024-11-22
> > **Response to Reviewer pnRJ (part2)**
> >
> > **Q4**: Can you provide more details on the stability of data prediction models at low NFEs, and when would noise prediction be preferable?
> >
> > **A**: Below, we present the results for NFE=5 on the inpainting task. Due to format limitations, some of the image results cannot be displayed here but will be included in the appendix in the revised version of the manuscript.
> >
> > |                     | PSNR  |   SSIM   |  LPIPS  |
> > | :--------------: | :---: | :------: | :-----: |
> > | noise prediction | 5.756 | 0.005541 | 0.9118  |
> > | data prediction  | 32.74 |  0.9517  | 0.03685 |
> >
> > Ratio of convergence at each time step w.r.t. different parameterizations.
> >
> > | Index of timestep |   1    |   2    |   3    |   4    |
> > | :---------------: | :----: | :----: | :----: | :----: |
> > | noise prediction  | 0.3431 | 0.9246 | 0.9992 | 0.9999 |
> > |  data prediction  | 0.2937 | 0.0380 | 0.0422 | 0.1115 |
> >
> > When the NFE is large, the performance of noise prediction closely aligns with that of data prediction. However, when the NFE is small, noise prediction proves to be inferior to data prediction. In summary, there are no scenarios in which noise prediction outperforms data prediction.
> >
> > **Q5**: Can MR Sampler be extended to larger or multi-modal tasks, such as video generation or text-to-image models?
> >
> > **A**: The MR Sampler is an efficient sampling algorithm specifically developed for MR Diffusion. MR Diffusion can be considered an extension of DPM based on its corresponding SDE. With the remarkable success of current DPM-based multi-modal large models, we believe that MR Diffusion possesses comparable or even greater potential. We aim to explore and validate this belief in follow-up studies.
> >
> > **Q1(b)&Q6&W2**: Comparisons with CCDF and SOTA methods in terms of both speed and quality
> >
> > **A**: To the best of our knowledge, existing sampling acceleration methods are specifically designed for DPM and our method is the first sampling acceleration algorithm for MR Diffusion. Numerous algorithms have achieved promising results in accelerating DPMs, including Come-Closer-Diffuse-Faster. Specifically, Come-Closer-Diffuse-Faster is designed for DPM (VP-SDE and VE-SDE). The corresponding SDE for DPM is given as $\mathrm{d}\boldsymbol{x}=f(t)\boldsymbol{x}\mathrm{d}t + g(t) \mathrm{d}\boldsymbol{w} \;(1)$, whereas the SDE for MR Diffusion is $\mathrm{d}\boldsymbol{x}=f(t)\left(\boldsymbol{\mu}-\boldsymbol{x}\right)\mathrm{d}t+g(t)\mathrm{d}\boldsymbol{w} \;(2)$. When $\boldsymbol\mu=0$, it can be shown that Equation (2) reduces to Equation (1). This implies that DPM is a special case of MR Diffusion, and MR Diffusion can be regarded as a generalization of DPM. Consequently, existing fast sampling methods designed for DPM are not directly applicable to MR Diffusion. Following the thoughtful suggestion of the reviewer, we have added a discussion of these methods in the background section of the revised manuscript.

---

### Official Review · Reviewer_wsmd · 2024-11-01

**Soundness:** 3
**Presentation:** 3
**Contribution:** 3
**Rating:** 8
**Confidence:** 3

**Summary:**

The authors propose a novel algorithm, named MRS, designed as a sampler to reduce the number of sampling steps required for Mean Reverting Diffusion. To achieve this, they solve the reverse-time stochastic differential equation (SDE) alongside the probability flow ordinary differential equation, resulting in a reduction of required steps to between 5 and 15 while maintaining high-quality outcomes. The method is compatible with mainstream parameterizations, making it broadly applicable. The authors conduct an extensive evaluation across 16 different image perturbations, demonstrating that their proposed sampler generally outperforms others in both efficiency and speed.

**Strengths:**

- The authors' proposed method demonstrates robust performance across a variety of experiments, consistently showing strong results in both quality and efficiency. Notably, it outperforms posterior-based and Euler-based samplers when using a high number of sampling steps. The performance advantage becomes even more pronounced as the number of steps is reduced, underscoring the method's effectiveness with low-step counts.

- It is particularly encouraging to see that the proposed method performs best across almost all of the 16 tested image perturbations. This breadth of performance suggests that the approach is not only effective but also adaptable to different types of image perturbations.

- The paper is generally well-organized and written. Key concepts and technical choices are clearly explained.

- Overall, this is a well-rounded paper that offers convincing results and provides a method that should be straightforward to integrate with existing frameworks. I also appreciate the authors’ decision to release the code for easy reproducibility.

**Weaknesses:**

- The choice of the number of sampling steps is not entirely clear. While it seems that setting this value to 20 yields acceptable results, this may not fully leverage the acceleration benefits of the proposed method. Further guidance on selecting an optimal number of steps or a discussion on its trade-offs would make this aspect more transparent.
- There is some ambiguity regarding the number of function evaluations (NFEs) required for MR Diffusion. The authors mention that MR Diffusion typically requires hundreds of NFEs, but in the current paper, this value is set to 100, consistent with the original paper. This raises questions about whether the maximum performance of the Posterior and Euler-based sampling methods could be higher if a larger number of steps were used. Given the performance gains depicted in Figure 2, it would be helpful to clarify if the efficiency improvements observed are partly due to these optimized sampling steps.
- Table 15 in the appendix contains incorrect highlight

**Questions:**

- How did you determine the optimal number of sampling steps? The choice of steps varies across different experiments, but the selection criteria are not immediately clear. Could you provide more explanation on how to select the number of steps to balance efficiency and performance?
- What was the rationale behind choosing a low-light and motion-blurry dataset for the visualizations in Figure 2?
- In reference to the related paper [1], it is mentioned that solving the SDE typically requires only 22 steps, although they still use 100 steps. Since you opted to use 100 steps aswell, I am wondering why the performance with 20 steps is still so substantially different.

[1] Luo, Ziwei, et al. "Image restoration with mean-reverting stochastic differential equations." arXiv preprint arXiv:2301.11699 (2023).

---

> ### Author Response · Authors · 2024-11-22
> **Response to Reviewer wsmd (part1)**
>
> Thank you for the interest and acknowledgment of our contributions and the insightful questions. Below we respond to the weaknesses and questions.
>
> **Q1&W1**: How to determine the optimal number of sampling steps
>
> **A**: The NFE is strictly positively correlated with sampling quality. From the perspective of reverse-time SDE and PF-ODE, the estimation error of the solution plays a critical role in determining the sampling quality. This estimation error is primarily influenced by $h_{max} = \max_{1 \leq i \leq M}{\lambda_i - \lambda_{i-1}} = \mathcal{O}(\frac{1}{M})$, where $M$ represents the NFE. During sampling, the noise schedule must be designed in advance, which also determines the $\lambda$ schedule. Since $\lambda_i$ is monotonically increasing, a larger NFE results in a smaller $h_{max}$, thereby reducing the estimation error and improving sampling quality.
>
> However, different sampling algorithms exhibit varying convergence rates. Experimental results show that the MR Sampler (our method) can achieve a good score with as few as 5 NFEs, whereas posterior sampling and Euler discretization fail to converge even with 50 NFEs.
>
> Specifically, we conducted additional experiments on the desnow task with low NFE settings, and the results are presented in Table below. The sampling quality remains largely stable for NFE values larger than 10, gradually deteriorates when the NFE is below 10, and collapses entirely when NFE is reduced to 2. Based on our experience, we recommend using 10–20 NFEs, which provides a reasonable trade-off between efficiency and performance. We have added a section to appendix to provide an in-depth discussion on determining the NFE.
>
> MR Sampler-SDE-2 with data prediction and uniform $\lambda$ on the desnow task
>
> | NFE  | LPIPS  |  FID  | PSNR  |  SSIM  |
> | :--: | :----: | :---: | :---: | :----: |
> | 100  | 0.0626 | 21.47 | 27.18 | 0.8691 |
> |  50  | 0.0628 | 21.85 | 27.08 | 0.8685 |
> |  20  | 0.0650 | 22.34 | 26.89 | 0.8645 |
> |  10  | 0.0698 | 23.92 | 26.49 | 0.8573 |
> |  8   | 0.0725 | 24.81 | 26.61 | 0.8462 |
> |  6   | 0.0744 | 25.60 | 26.40 | 0.8407 |
> |  5   | 0.0628 | 21.95 | 27.06 | 0.8718 |
> |  4   | 0.1063 | 30.18 | 25.38 | 0.7640 |
> |  2   | 1.422  | 421.1 | 6.753 | 0.0311 |
>
> **Q2**: What was the rationale behind choosing a low-light and motion-blurry dataset for the visualizations in Figure 2?
>
> **A**: We conducted extensive experiments on ten datasets. Due to the page limitations of the main text, it was not feasible to include results for all datasets. As a result, we randomly selected two datasets to serve as representatives, and the results for the remaining datasets are provided in Appendix D2.
>
> **Q3**: In reference to the related paper [1], it is mentioned that solving the SDE typically requires only 22 steps, although they still use 100 steps. Since you opted to use 100 steps aswell, I am wondering why the performance with 20 steps is still so substantially different.
>
> **A**: We would like to emphasize that in paper [1], the authors were able to achieve sampling with 22 NFEs only for denoising tasks. In the sampling process of MR Diffusion, the initial state is $\mu + \sigma\epsilon$, where $\mu$ is typically set to a low-quality image, and the final state is $\boldsymbol{x}_0$, which is usually a high-quality image. For denoising tasks, the authors assume that the low-quality image is equivalent to the high-quality image plus Gaussian noise, implying $\boldsymbol{x}_0 = \mu$ (see Section 4.3 and Appendix B in paper [1]). This leads to $\boldsymbol{x}\_t = \mu + \sigma\epsilon = \boldsymbol{x}\_0 + \sigma\epsilon$. In this case, the PF-ODE simplifies to $\mathrm{d}\boldsymbol{x}=-\frac12\boldsymbol{\sigma}^2\_t\mathrm{e}^{-2\bar\theta\_t}\nabla\_{x}\log{p\_t(\boldsymbol{x})}\mathrm{d}t.(1)$
>
> It is important to note that this scenario is equivalent to setting $f(x,t) = 0$, causing the linear term in the PF-ODE to cancel out. In our paper, we explain in Section 3.3 the reasons for the slow convergence of Euler discretization, which introduces approximation errors from both linear and nonlinear terms. However, for Equation (1), the absence of the linear term eliminates the corresponding error, allowing for the use of a smaller NFE in this specific case. Such simplification may not hold for other tasks because they require the degradation of the image to result exclusively from additive Gaussian noise. In contrast, our sampling algorithm is not subject to this limitation, making it more broadly applicable.
>
> [1]. Ziwei Luo, Fredrik K Gustafsson, Zheng Zhao, Jens Sjölund, and Thomas B Schön. Image restoration with mean-reverting stochastic differential equations. arXiv preprint arXiv:2301.11699, 2023b.

---

> > ### Author Response · Authors · 2024-11-22
> > **Response to Reviewer wsmd (part2)**
> >
> > **W2**: The reason why NFE is set to 100 and the question about whether performance could improve with a larger NFE.
> >
> > **A**: We regret the confusion caused by our writing regarding NFE. In the introduction, we mention that "MR Diffusion requires hundreds of iterative steps." However, we did not highlight an important premise here. Since DDPM was proposed, $N=1k$ has commonly been used during the training phase of diffusion models. As a result, the upper bound of NFE during sampling for such models is also typically set to $1k$.
> >
> > In contrast, Mean Reverting Diffusion [1] employs $N=100$ during its training, making the upper bound of NFE during sampling for MR Diffusion only 100. Conceptually, the diffusion model learns to fit the score function during training, effectively teaching the neural network how to denoise across different noise levels. From this perspective, using a larger $N$ enables the neural network to learn noise levels with greater precision, potentially improving the model's performance during sampling.
> >
> > We speculate that the decision to set $N=100$ in [1] was driven by a desire to minimize sampling time. The authors provided open-source code and pre-trained checkpoints, which we directly utilized for convenience. Consequently, we were restricted to sampling with a maximum NFE of 100. Due to time constraints, it is not feasible for us to train the model for $N>100$ steps, which we plan to explore in future work. Nevertheless, if $N=1k$ were used during the training phase, our sampling method would still converge in approximately 20 NFEs.
> >
> > **W3**: Table 15 in the appendix contains incorrect highlight.
> >
> > **A**: We regret the error and have corrected it in the revised manuscript.

---

> > > ### Comment · Reviewer_wsmd · 2024-11-27
> > >
> > > Thank you for the additional clarifications. I keep my original score as this is a good paper and should be accepted.

---

### Official Review · Reviewer_hx57 · 2024-11-04

**Soundness:** 3
**Presentation:** 3
**Contribution:** 3
**Rating:** 6
**Confidence:** 3

**Summary:**

This paper proposes a fast sampling algorithm for mean-reverting diffusion. This addresses a gap for mean-reverting diffusion SDE solvers, as current fast samplers for SDEs do not readily apply to mean-reverting SDEs. Two flavors of solvers are proposed, one based on noise prediction, and the other on data prediction. Results show both perform similarly for larger NFEs, but the latter outperforms the former for fewer NFEs.

**Strengths:**

- Addresses a gap in fast sampling for mean-reverting diffusion
- Provides two alternatives focusing on noise/data prediction
- Relevant ablation studies for some parameter choices
- Evaluates performance in image restoration tasks

**Weaknesses:**

- No ablation study on n, k
- Wall clock time not reported, only NFE improvement is discussed
- No comparison to standard SDE fast samplers
- The whole description is for unconditional sampling, but all results are for image restoration. How is the guidance incorporated?

**Questions:**

- Why wasn't an ablation study done on n, k?
- Why wasn't wall clock time reported? Please provide.
- Why weren't there comparison to standard (non-MR) SDE fast samplers? This comparison should include both reconstruction quality for inverse problems (since this is given as the main motivation for MR diffusion) and computational time.
- How was guidance incorporated for image restoration? Does the particular method for incorporating the degraded observation make a difference for the performance of MR sampler?

---

> ### Author Response · Authors · 2024-11-21
> **Response to Reviewer hx57**
>
> We thank the reviewer for the thoughtful and constructive comments. Below we provide detailed responses to the questions.
>
> **Q1**: Why wasn't an ablation study done on n, k?
>
> **A**: Because the variable $n$ is not explicitly defined or utilized in our paper. We assume that the reviewer uses $n$ to refer to the NFE (number of function evaluations) and $k$ as the order of the algorithm. We have performed ablation studies on values of NFE and $k$. Detailed results are shown in Figure 2 of the main text and Tables 6–15 in Appendix D2.
>
> **Q2**: Why wasn't wall clock time reported? Please provide.
>
> **A**: We regret not reporting the wall clock time in our original manuscript. We indeed measured it in our experiments on a single NVIDIA A800 GPU. The average wall clock time per image of two representative tasks (low-light and motion-blurry image restoration) are reported below, and we have added a section D.5 to the Appendix to report the wall clock time.
>
> | NFE  |     Low-light      | Time(s) |   Motion-blurry    | Time(s) |
> | :--: | :----------------: | :-----: | :----------------: | :-----: |
> | 100  | Posterior Sampling |  17.19  | Posterior Sampling |  82.04  |
> |  50  |                    |  8.605  |                    |  41.23  |
> |  20  |                    |  3.445  |                    |  16.44  |
> |  10  |                    |  1.727  |                    |  8.212  |
> |  5   |                    | 0.8696  |                    |  4.133  |
> | 100  |    MR Sampler-2    |  17.83  |    MR Sampler-2    |  81.16  |
> |  50  |                    |  8.439  |                    |  40.18  |
> |  20  |                    |  3.285  |                    |  15.59  |
> |  10  |                    |  1.569  |                    |  7.413  |
> |  5   |                    | 0.7112  |                    |  3.294  |
>
>
> As observed, the time consumption and NFE are approximately proportional. This relationship arises because the sampling time is predominantly determined by neural network inference, while the time required for other computations is negligible in comparison.
>
> **Q3**: comparison to standard (non-MR) SDE fast samplers
>
> **A**: To the best of our knowledge, existing sampling acceleration methods are specifically designed for DPM, so these methods are not directly applicable to MR Diffusion. Therefore, when comparing the MR Sampler and the standard SDE Sampler, the diffusion models used by the two methods are different. Specifically, the standard SDE Sampler is for the DPM, while the MR Sampler is for the MR Diffusion. However, to compare standard SDE sampler with MR Sampler is questionable, as DPM and MR Diffusion exhibit different levels of performance on the image restoration task. Under such conditions, it becomes difficult to discern whether the observed performance differences arise from the sampling algorithm itself or from the diffusion model.
>
> **Q4**: How was guidance incorporated for image restoration? Does the particular method for incorporating the degraded observation make a difference for the performance of MR sampler?
>
> **A**: MR Diffusion (MRD) is indeed a type of conditional diffusion model. Luo et al. [1] developed MRD by redesigning the drift coefficient of the SDE and incorporating $\mu$ into the diffusion model. In the diffusion process of DPM, the trajectory starts from $\boldsymbol{x}_0$ and converges to pure Gaussian noise. In contrast, the diffusion process of MRD begins at $\boldsymbol{x}_0$ and converges to $\mu + \sigma\epsilon$. The introduction of $\mu$ serves as a conditional input channel for MRD. For image-to-image tasks, MRD assigns the input image to $\mu$, eliminating the need to incorporate image conditions via Classifier-Free Guidance (CFG).
>
> Actually, incorporating $\mu$ does not make a favorable difference to the performance of MR Sampler. However, it is worth noting that the inclusion of $\mu$ increases the complexity of the reverse-time SDE and PF-ODE, making the solution of these equations more challenging.
>
> [1]. Ziwei Luo, Fredrik K Gustafsson, Zheng Zhao, Jens Sjölund, and Thomas B Schön. Image restoration with mean-reverting stochastic differential equations. arXiv preprint arXiv:2301.11699, 2023b.

---

> > ### Comment · Reviewer_hx57 · 2024-11-25
> >
> > I thank the reviewers for their revision. I am still unsure about Q3 - while I understand the confounding factor issue, as a diffusion model user, I'd want to know how fast MR sampling works versus non-MR fast sampling, as this would affect the choice of my "baseline" algorithm.
> >
> > Nonetheless, I have raised my score.
> >
> > I also want to note that I am not an expert on MR diffusion models, and have indicated that in my confidence score.

---

> > > ### Author Response · Authors · 2024-11-29
> > > **Author Responses**
> > >
> > > Thanks for your comments. Despite the presence of confounding factors, this is an important question. When it comes to generation tasks with image prompts, DDPM-based models incorporate image prompt information through neural networks. However, there is a potential risk of information loss when these networks extract image features. In contrast, MR Diffusion benefits from a natural image input channel and is capable of learning a mapping between two arbitrary distributions. Due to the limited time available for the rebuttal period, it is difficult to conduct experiments and verify our opinion. We are willing to explore this question further in future work.

---

### Official Review · Reviewer_q9n1 · 2024-11-04

**Soundness:** 4
**Presentation:** 4
**Contribution:** 3
**Rating:** 8
**Confidence:** 4

**Summary:**

Standard MR Diffusion models typically require a high number of function evaluations to obtain high-quality samples. To address this limitation, MR Sampler introduces a novel approach that combines an analytical function with an integral parameterized by a neural network. The proposed method demonstrates a 10 to 20-fold speedup across various image restoration tasks, obtaining efficient sampling without compromising quality.

**Strengths:**

1). The paper is very well written, and easy to understand especially very coherent even with the math part.

2). The core idea of the paper is very interesting. Especially how the authors have formulated  the core idea.

3). The shown results are very impressive and competitive.

4). Even though the concept behind the "sampling trajectory" part is simple, the visualizations obtained through it really helps to understand the core concept and contrast with existing methods.

5). The MRS is plug and play.

**Weaknesses:**

I believe this is a great paper with solid reasoning and a well-thought-out approach. However, I have some questions regarding the numerical implementation and the comparison methods used.

1). Can the authors elaborate on why only the backward difference method is used? Are there specific benefits to this approach over others?

2). The proposed MR Sampler is only compared with posterior sampling, and Euler-Maruyama discretization. What are the current SOTA methods? How does the proposed approach compare with these, beyond posterior sampling and Euler-Maruyama?

3). I would like to understand why the authors say "frequently fall outside" in this part (line304)
 "Although the standard deviation of this Gaussian noise is set to 1, the values of samples can frequently fall outside the range of [-1,1]"

4). Could the authors specify the neural network architecture they used?

Please address these, and I'm leaning towards acceptance.

**Questions:**

Please see the weaknesses section.

---

> ### Author Response · Authors · 2024-11-21
> **Response to Reviewer q9n1**
>
> We thank the reviewer for the thoughtful and constructive comments. Below we provide detailed responses to the questions.
>
> **W1**: reason for using the backward difference method
>
> **A**: We use the backward difference method to estimate the derivative of the neural network. Here we take Equation (14) from our paper as an example.
> $$
> -2\alpha\_t\int\_{\lambda_s}^{\lambda_t}e^{-\lambda}\boldsymbol{\epsilon}\_\theta(x\_\lambda,\lambda)\mathrm{d}\lambda=
> -2\sigma\_{t}\sum\_{n=0}^{k-1}\left[\boldsymbol{\epsilon}\_\theta^{(n)}(\boldsymbol{x}\_{\lambda_{s}},\lambda_{s})\left(e^{h}-\sum\_{m=0}^n\frac{(h)^m}{m!}\right)\right]
> +\mathcal{O}(h^{k+1}) \tag{14}
> $$
> Equation (14) represents the non-linear component of the solution presented in Equation (13). Our objective is to estimate the derivative of the neural network $\boldsymbol{\epsilon}^{(n)}\_\theta(\boldsymbol{x}\_{\lambda},\lambda)$ with respect to the signal-to-noise ratio (SNR) $\lambda$. Using $\boldsymbol{\epsilon}^{(1)}\_\theta(\boldsymbol{x}\_{\lambda},\lambda)$ as a specific example, as illustrated in Algorithms 1–6, this process involves discretizing time and computing the solution iteratively. Specifically, we need to estimate $\boldsymbol{\epsilon}^{(1)}\_\theta(\boldsymbol{x}\_{\lambda_i},\lambda_i)$ at each time step. Because the analytical expression of $\boldsymbol{\epsilon}\_\theta$ is unavailable, the derivative is commonly approximated by the *difference method*, including forward difference, central difference, and backward difference approaches. However, the forward and central difference methods require the value of $\boldsymbol{\epsilon}\_\theta(\boldsymbol{x}\_{\lambda_{i+1}},\lambda_{i+1})$, but $\boldsymbol{x}\_{\lambda_{i+1}}$ itself needs to be predicted. Consequently, we adopt the backward difference method in this case.
>
> **W2**: current SOTA methods
>
> **A**: To the best of our knowledge, existing sampling acceleration methods are specifically designed for DPM, and our method is the first sampling acceleration algorithm for MR Diffusion. The corresponding SDE for DPM is given as $\mathrm{d}\boldsymbol{x}=f(t)\boldsymbol{x}\mathrm{d}t + g(t) \mathrm{d}\boldsymbol{w} \;(1)$, whereas the SDE for MR Diffusion is $\mathrm{d}\boldsymbol{x}=f(t)\left(\boldsymbol{\mu}-\boldsymbol{x}\right)\mathrm{d}t+g(t)\mathrm{d}\boldsymbol{w} \;(2)$. When $\boldsymbol\mu=0$, it can be shown that Equation (2) reduces to Equation (1). This implies that DPM is a special case of MR Diffusion, and MR Diffusion can be regarded as a generalization of DPM. Consequently, existing fast sampling methods designed for DPM are not directly applicable to MR Diffusion.
>
> **W3**: "frequently fall outside" in line304
>
> **A**: The original text is "During the training phase, the noise prediction neural network is designed to fit normally distributed Gaussian noise. Although the standard deviation of this Gaussian noise is set to 1, the values of samples can frequently fall outside the range of [-1, 1]."
> We apologize for any confusion here. The original statement pertains to the $3\sigma$ rule of the Gaussian distribution. Specifically, for a random variable $x$ that follows a normal distribution, the probability of $x \in [-1, 1]$ is approximately 65.26%, while the probability of $x \notin [-1, 1]$ is 34.74%. We acknowledge that the use of *frequently* in this context is not rigorous. We will revise the text as follows: "When the standard deviation of this Gaussian noise is set to 1, the values of samples can fall outside the range of [-1,1] with a probability of 34.74%."
>
> **W4**: the neural network architecture used in experiments
>
> **A**: We sincerely apologize for not explicitly clarifying the model framework and neural network employed in our experiments. Our work fully adheres to the framework of DA-CLIP [1], an image restoration model designed to address multiple degradation problems simultaneously without requiring prior knowledge of the degradation. The diffusion model in DA-CLIP is derived from Refusion [2], and its neural network architecture is based on NAFNet. NAFNet builds upon the U-Net architecture by replacing traditional activation functions with SimpleGate and incorporating an additional multi-layer perceptron to manage channel scaling and offset parameters for embedding temporal information into the attention and feedforward layers. For further details, please refer to Section 4.2 in [2]. We have added a section D.2 to the Appendix to describe the network architecture used in experiments.
>
> [1]. Ziwei Luo, Fredrik K Gustafsson, Zheng Zhao, Jens Sjölund, and Thomas B Schön. Controlling vision-language models for multi-task image restoration. In The Twelfth International Conference on Learning Representations, 2024a.
>
> [2]. Ziwei Luo et al. Refusion: Enabling large-size realistic image restoration with latent-space diffusion models. In Proceedings of the IEEE/CVF conference on computer vision and pattern recognition, pp. 1680–1691, 2023c.

---

> > ### Comment · Reviewer_q9n1 · 2024-11-25
> >
> > Thank you very much for the response. All of my concerns have been addressed, and I raised my ratings.

---

### Author Response · Authors · 2024-11-22
**Summary of Revisions**

We really appreciate the thoughtful and constructive comments of all reviewers. We have tried our best to revise the paper to address all concerns.  All changes have been highlighted in purple for clarity. Specifically:

- **@Reviewer1 q9n1**, We have added a section D.2 to the Appendix to describe the network architecture used in experiments.
- **@Reviewer pnRJ**, we have presented further details on the inpainting task for NFE=5 in Appendix D.4.
- **@Reviewer hx57**, we have reported the average wall clock time per image on two representative tasks (low-light and motion-blurry image restoration) in Appendix D.5.
- **@Reviewer wsmd** and **Reviewer pnRJ**, we have provided an in-depth discussion on determining the optimal NFE in Appendix E.

Please let us know if you have any further questions. We welcome continued discussions and are always open to feedback.

Sincerely,

Authors

---

### Meta-Review · Area_Chair_Urun · 2024-12-22

**Metareview:**

The paper introduces MR Sampler, a novel algorithm designed to accelerate sampling in Mean Reverting Diffusion models by combining semi-analytical solutions from ordinary differential equations (ODE) and stochastic differential equations (SDE). This approach addresses the inefficiency of previous MR diffusion models, reducing the required sampling steps while maintaining high-quality outputs. MR Sampler demonstrates a 10 to 20-fold speedup across 16 image restoration tasks, providing robust and competitive results. The method is well-supported by theoretical grounding, integrates seamlessly with existing frameworks, and bridges a critical gap in fast sampling for MR diffusion models. However, the reviewers raise the questions about how to determine the optimal sampling step, wall-clock time reporting, and baseline comparison. Besides, the paper may be limited by mainly focusing on the speed of sampling rather than the generation quality. In the rebuttal, the author has provided a comprehensive response for clarification and almost all of the concerns have been well addressed. Thus, I recommend an acceptance.

**Additional Comments On Reviewer Discussion:**

All the reviewers have actively involved in the discussion with the author during the rebuttal period. After the discussion, quite a few reviewers raise the score to reflect that their concerns have been well addressed.

---

### Decision · Program_Chairs · 2025-01-22

Accept (Spotlight)